# A Dynamical System View of Langevin-Based Non-Convex Sampling

**Mohammad Reza Karimi**\*
ETH Zürich
mkarimi@inf.ethz.ch

**Ya-Ping Hsieh**\*
ETH Zürich
yaping.hsieh@inf.ethz.ch

**Andreas Krause**
ETH Zürich
krausea@ethz.ch

## Abstract

Non-convex sampling is a key challenge in machine learning, central to non-convex optimization in deep learning as well as to approximate probabilistic inference. Despite its significance, theoretically there remain some important challenges: Existing guarantees suffer from the drawback of lacking guarantees for the *last-iterates*, and little is known beyond the elementary schemes of stochastic gradient Langevin dynamics. To address these issues, we develop a novel framework that lifts the above issues by harnessing several tools from the theory of dynamical systems. Our key result is that, for a large class of state-of-the-art sampling schemes, their last-iterate convergence in Wasserstein distances can be reduced to the study of their continuous-time counterparts, which is much better understood. Coupled with standard assumptions of MCMC sampling, our theory immediately yields the last-iterate Wasserstein convergence of many advanced sampling schemes such as mirror Langevin, proximal, randomized mid-point, and Runge-Kutta methods.

## 1 Introduction

Many modern learning tasks involve sampling from a high-dimensional density $\pi \propto e^{-f}$, where $f$ is a *non-convex* potential representing, for instance, the loss function of a deep neural network. To this end, an approach that has found wide success is to discretize the continuous-time *Langevin diffusion*

$$\mathrm{d}L_t = -\nabla f(L_t)\,\mathrm{d}t + \sqrt{2}\,\mathrm{d}B_t \tag{LD}$$

where $B_t$ is a Brownian motion [57]. The idea behind this approach is that, since $\pi$ is the stationary distribution of (LD), one can expect a similar behavior for discretizations of (LD). Such a framework has inspired numerous sampling schemes with per-iteration costs as cheap as stochastic gradient descent, which are particularly suitable for large-scale approximate probabilistic inference and Bayesian learning [2, 54, 57]. Moreover, several works have noticed that these Langevin-based schemes provide deep insights about minimizing $f$ using stochastic oracles [22, 48], which serves as an important step toward explaining the empirical success of training deep neural networks.

The convergence of Langevin-based non-convex sampling has therefore attracted significant interest from both practitioners and theoreticians, whose intense study has led to a plethora of new guarantees; see related work for details. Despite such impressive progress, several challenges remain for the fully non-convex setup:

- The convergence is typically given on the *averaged* iterates instead of the more natural *last* iterates [4, 54]. This is especially problematic from the perspective of understanding the minimization of $f$, as in practice, the last iterates of an optimization algorithm play the most pivotal role for downstream tasks.

---

\*Equal contribution.

37th Conference on Neural Information Processing Systems (NeurIPS 2023).

- An additional notable drawback of the current theory is its predominant focus on the basic Euler-Maruyama discretization of (LD) (see, e.g., [4, 20, 54]). As a result, the convergence analysis of more advanced sampling schemes remains largely unexplored in the fully non-convex regime [1, 24, 25, 34, 53, 61].

§ **Contributions and Approaches.** To overcome the aforementioned challenges, our main contribution, from a high level, can be succinctly summarized as:

> Under mild assumptions, we prove that the iterates of a broad range of Langevin-based sampling schemes converge to the *continuous-time* (LD) in *Wasserstein distance*.  $(\star)$

Combining $(\star)$ with classical results on Langevin diffusion [45] immediately yields the *last-iterate* convergence in *Wasserstein distances* for a wide spectrum of sampling schemes, thus resolving all the challenges mentioned above. To illustrate this point, we state a simple version of our main result.

**Theorem** (Informal). *Suppose we discretize* (LD) *as*

$$x_{k+1} = x_k - \gamma_{k+1}(\nabla f(x_k) + noise + bias) + \sqrt{2\gamma_{k+1}}\,\xi_{k+1}$$

*with step-sizes* $\{\gamma_k\}_{k \in \mathbb{N}}$ *and i.i.d. standard Gaussians* $\{\xi_k\}_{k \in \mathbb{N}}$. *Then, under an easy-to-verify condition on the bias (see* (5) *in Assumption 3),* $\{x_k\}_{k \in \mathbb{N}}$ *converges in Wasserstein distance to* $\pi$. *In addition, these conditions are satisfied by many advanced sampling schemes.*

This result is achieved via a new *dynamical perspective* to study Langevin-based sampling. More specifically,

1. We introduce the *Picard process*, which is the sampling analogue of Picard's method of successive approximations for solving ODEs [17]. Contrary to most existing analyses, the Picard process allows us to completely bypass the use of relative entropy, which is the culprit for the appearance of averaged iterates [20].

2. Using the Picard process, we will prove that the iterates of various Langevin-based schemes generate a so-called *Wasserstein asymptotic pseudotrajectory* (WAPT) for the continuous-time (LD). The main motivation for considering WAPT is to connect Langevin-based schemes to the dynamical system theory of Benaïm and Hirsch [7], which works for metric spaces and is last-iterate by design, and therefore particularly suitable for our purpose.

3. Finally, under standard stability assumptions in the literature [39, 51], we show how a tandem of our WAPT result and dynamical system theory yields the desirable convergence of various existing schemes, as well as motivates more efficient algorithms that enjoy the same rigorous guarantees.

§ **Related work.** There is a vast literature on *structured* non-convex sampling, where one imposes extra assumptions on the target density. Under these conditions, one can derive *non-asymptotic* rates for Langevin-based schemes [13, 15, 33, 35, 36, 41, 48, 56, 60, 63]. Our work is orthogonal to these works as we study *generic* non-convex sampling, an NP-hard problem whose convergence is asymptotic at best.

Most relevant to our paper are the works [4, 8, 20, 29, 54], which study the asymptotic convergence of Langevin-based schemes under minimal regularity assumptions on $f$. Compared to their results, our guarantees either improve upon existing ones or are incomparable; see Section 5.4 for a more detailed comparison.

## 2 The Langevin-Robbins-Monro Template

We consider the following general template for sampling algorithms: Starting from an initial point, the iterates $\{x_k\}_{k \in \mathbb{N}}$ follow the recursion

$$x_{k+1} = x_k - \gamma_{k+1}\{v(x_k) + Z_{k+1}\} + \sqrt{2\gamma_{k+1}}\,\sigma(x_k)\,\xi_{k+1}, \tag{LRM}$$

where $\gamma_k$'s are step sizes, $v$ is a vector field, $Z_k$'s are (random or deterministic) perturbations, $\sigma$ is the state-dependent diffusion matrix, and $\xi_k$'s are i.i.d. standard Gaussian random variables. In the sequel, we will further decompose the perturbation as $Z_k = U_k + b_k$, where $U_k$ is the (zero-mean)

*noise* and $b_k$ is the *bias*. We call this recursion the *Langevin-Robbins-Monro* (LRM) template, as it is reminiscent of the Robbins-Monro template for stochastic approximation [50].

The generality of the LRM template allows us to capture many existing algorithms and suggests ways to design new ones. For illustration purposes, we showcase instances of (LRM) with the following examples. Other examples (SGLD and proximal) are provided in Appendix A. In the first three examples, the vector field $v$ in (LRM) is $-\nabla f$ and $\sigma \equiv 1$.

**Example 1.** The *Randomized Mid-Point Method* [24, 53] is an alternative discretization scheme to Euler-Maruyama and has been proposed for both overdamped and underdamped Langevin diffusion. For the overdamped case, its iterates are

$$x_{k+1/2} = x_k - \gamma_{k+1}\alpha_{k+1}\widetilde{\nabla}f(x_k) + \sqrt{2\gamma_{k+1}\alpha_{k+1}}\,\xi'_{k+1},$$
$$x_{k+1} = x_k - \gamma_{k+1}\widetilde{\nabla}f(x_{k+1/2}) + \sqrt{2\gamma_{k+1}}\,\xi_{k+1}, \qquad \text{(RMM)}$$

where $\{\alpha_k\}$ are i.i.d. and uniformly distributed in $[0, 1]$, $\xi_k, \xi'_k$ are standard Gaussian random variables with cross-variance $\sqrt{\alpha_k}I$, and $\widetilde{\nabla}f$ is a noisy evaluation of $\nabla f$. To cast (RMM) in the LRM template, we set $U_{k+1} := \widetilde{\nabla}f(x_{k+1/2}) - \nabla f(x_{k+1/2})$ and $b_{k+1} := \nabla f(x_{k+1/2}) - \nabla f(x_k)$. □

**Example 2.** Inspecting the update rule of (RMM), we see that it requires *two* gradient oracle calls at each iteration. Inspired by the *optimistic gradient methods* in optimization and online learning [16, 47, 49], we propose to "recycle" the past gradients:

$$x_{k+1/2} = x_k - \gamma_{k+1}\alpha_{k+1}\widetilde{\nabla}f(x_{k-1/2}) + \sqrt{2\gamma_{k+1}\alpha_{k+1}}\,\xi'_{k+1},$$
$$x_{k+1} = x_k - \gamma_{k+1}\widetilde{\nabla}f(x_{k+1/2}) + \sqrt{2\gamma_{k+1}}\,\xi_{k+1}, \qquad \text{(ORMM)}$$

where $\{\alpha_k\}, \xi_k, \xi'_k$, and $\widetilde{\nabla}f$ are the same as in (RMM). This is again an LRM scheme with $U_{k+1} := \widetilde{\nabla}f(x_{k+1/2}) - \nabla f(x_{k+1/2})$ and $b_{k+1} := \nabla f(x_{k+1/2}) - \nabla f(x_k)$.

Notice that (ORMM) requires *one* gradient oracle, thereby reducing the per-iteration cost of (RMM) by 2. To our knowledge, the scheme (ORMM) is new. □

**Example 3.** In addition to the simple (stochastic) Euler-Maruyama discretization in (SGLD), there exists a class of more sophisticated discretization methods of (LD) known as higher-order integrators. The *Stochastic Runge-Kutta method* [34] is an example of an order 1.5 integrator, with iterates

$$h_1 = x_k + \sqrt{2\gamma_{k+1}}\,(c_1\xi_{k+1} + c_2\xi'_{k+1})$$
$$h_2 = x_k - \gamma_{k+1}\widetilde{\nabla}f(x_k) + \sqrt{2\gamma_{k+1}}\,(c_3\xi_{k+1} + c_2\xi'_{k+1}),$$
$$x_{k+1} = x_k - \frac{\gamma_{k+1}}{2}(\widetilde{\nabla}f(h_1) + \widetilde{\nabla}f(h_2)) + \sqrt{2\gamma_{k+1}}\,\xi_{k+1},$$

where $\xi_{k+1}$ and $\xi'_{k+1}$ are independent standard Gaussian random variables, and $c_1, c_2, c_3$ are suitably chosen integrator constants. This algorithm is an LRM scheme with $U_{k+1} := \frac{1}{2}(\widetilde{\nabla}f(h_1) - \nabla f(h_1)) + \frac{1}{2}(\widetilde{\nabla}f(h_2)) - \nabla f(h_2))$ and $b_{k+1} := \frac{1}{2}(\nabla f(h_1) + \nabla f(h_2)) - \nabla f(x_k)$.

**Example 4.** The *Mirror Langevin* algorithm [1, 25, 61], which is the sampling analogue of the celebrated mirror descent scheme in optimization [5, 43], uses a strongly convex function $\phi$ to adapt to a favorable local geometry. In the dual space (i.e., the image of $\nabla\phi$), its iterates follow

$$x_{k+1} = x_k - \gamma_{k+1}\nabla f(\nabla\phi^*(x_k)) + \sqrt{2\gamma_{k+1}}(\nabla^2\phi^*(x_k)^{-1})^{1/2}\,\xi_{k+1}, \qquad \text{(ML)}$$

where $\phi^*$ is the *Fenchel dual* of $\phi$ [52]. In our framework, (ML) fits into (LRM) by taking $v = -\nabla f \circ \nabla\phi^*$ and $\sigma = (\nabla^2\phi^*)^{-1/2}$. Additionally, one can also consider a stochastic version of (ML) with noisy evaluations of $\nabla f$.

## 3 Technique Overview: A Dynamical System Perspective

The goal of our paper is to provide last-iterate guarantees for the general LRM schemes introduced in Section 2. There are two equivalent, commonly considered, ways of characterizing the dynamics of the iterates of an LRM scheme. The first one is to view the iterates $\{x_k\}_{k\in\mathbb{N}}$ as a *random* trajectory in $\mathbb{R}^d$, which is perhaps the most natural way of describing a sampling algorithm. The second way is to view the *distributions* $\{\rho_k\}_{k\in\mathbb{N}}$ of $\{x_k\}_{k\in\mathbb{N}}$ as a *deterministic* trajectory in the *Wasserstein space*.

With these two characterizations in mind, in this section, we will devise a new framework based on the dynamical system theory and present its high-level ideas.

To understand our novelty, it is important to contrast our framework to the existing Wasserstein viewpoint towards Langevin-based sampling algorithms. Following the seminal work of Otto [44], one can view a sampling algorithm as the discretization of a class of well-studied dynamical systems—*gradient flows*. This viewpoint suggests using *Lyapunov* arguments, which has become the predominant approach in much prior work.

Despite its appealing nature, in the rest of this section, we will argue that Lyapunov analysis of gradient flows is in fact *not* suited for studying generic non-convex sampling. In particular, we will show how our new framework is motivated to overcome the several important limitations of gradient flow analysis. Finally, we give a high-level overview of the techniques used in our paper.

**§ Langevin Diffusion as Gradient Flows.** We denote by $\rho_t$ the probability density of $L_t$ in (LD), and consider the continuous curve $t \mapsto \rho_t$ in the Wasserstein space $\mathbb{W}_2$. In their seminal works, Jordan et al. [27] and Otto [44] discover that this curve is the (exact) gradient flow of the relative entropy functional; that is, defining the functional $F : \rho \mapsto D_{\mathrm{KL}}(\rho \| e^{-f})$, one has $\partial_t \rho_t = -\operatorname{grad} F(\rho_t)$, where "grad" is the gradient in the Wasserstein sense. This gradient flow viewpoint of (LD) thus provides a clear link between sampling in $\mathbb{R}^d$ and optimization in $\mathbb{W}_2$. Indeed, this suggests that the relative entropy is a natural choice for the *Lyapunov function* of the discrete-time sampling algorithm, which is a prominent approach for analyzing sampling algorithms in recent years [4, 21, 58].

Although the gradient flow viewpoint has led to a sequence of breakthroughs, it has a few important shortcomings:

(a) The usual Lyapunov-type analysis for sampling algorithms focuses on bounding the change in relative entropy across iterations. This is extremely challenging when one considers more advanced sampling algorithms, as one has to understand the effect of the additive bias and noise of the algorithm on the change of relative entropy. Crucially, this makes the Lyapunov analysis applicable only to the simple Euler-Maruyama discretization of (LD),[2] i.e., $x_{k+1} = x_k - \gamma_{k+1} \nabla f(x_k) + \sqrt{2\gamma_{k+1}} \xi_{k+1}$, and fails to capture more advanced and *biased* sampling schemes such as Examples 1–4. Even for the simple (SGLD), the presence of stochastic gradients significantly complicates the Lyapunov analysis and requires extra assumptions such as convexity [21] or uniform spectral gap [48].

(b) This gradient flow-based analysis often requires an extra *averaging* step to decrease the relative entropy (see, e.g., [4]). This is the main reason why many existing works provide guarantees only on the *averaged* iterates ($\bar{\rho}_k := \frac{1}{k} \sum_{i=1}^{k} \rho_i$) instead of the last ones ($\rho_k$).

In this paper, we overcome these limitations by introducing a new perspective, whose two ingredients are as follows.

**§ Wasserstein Asymptotic Pseudotrajectories.** A notion that will play a pivotal role in our analysis is the *Wasserstein asymptotic pseudotrajectory* (WAPT), which is a measure of "asymptotic closeness" in the Wasserstein sense, originally defined by Benaïm and Hirsch [7] for metric spaces:

**Definition 1** (Wasserstein asymptotic pseudotrajectory). We say the stochastic process $(X_t)_{t \geq 0}$ is a *Wasserstein asymptotic pseudotrajectory* (WAPT) of the SDE

$$\mathrm{d}\Phi_t = v(\Phi_t)\,\mathrm{d}t + \sigma(\Phi_t)\,\mathrm{d}B_t \qquad\qquad \text{(SDE)}$$

if, for all $T > 0$,

$$\lim_{t \to \infty} \sup_{0 \leq s \leq T} W_2(X_{t+s}, \Phi_s^{(t)}) = 0. \qquad\qquad (1)$$

Here, $\Phi_s^{(t)}$ is the solution of the SDE at time $s$ initialized at $X_t$, and $W_2$ is the 2-Wasserstein distance.

Despite the seemingly convoluted definition, WAPT can be intuitively understood as follows: Let $\{x_k\}_{k \in \mathbb{N}}$ be the iterates of a sampling scheme. Then, (1) simply posits that for sufficiently large

---

[2]While the Lyapunov-type analysis has been applied to elementary (i.e., unbiased) discretization schemes for other SDEs, such as the under-damped (i.e., kinetic) Langevin dynamics [19], our primary focus in this paper remains centered on the over-damped Langevin diffusion and similar SDEs.

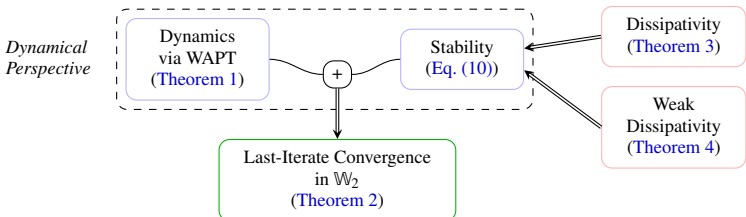

Figure 1: High-level overview of the two components of the dynamical perspective.

$m$, one cannot distinguish between the "tail" iterates $\{x_k\}_{k \geq m}$ versus the SDE solution *starting at* $x_m$, up to arbitrarily small error measured in terms of the Wasserstein distance. Since we are only interested in the asymptotic behavior of $x_k$, these controls on the tail iterates will suffice to conclude the last-iterate convergence.[3]

Importantly, from the perspective of WAPT, the Langevin diffusion (LD) (or more generally, $\Phi_s^{(t)}$) is simply viewed as a generic dynamical system and *not* as a gradient flow. In particular, *relative entropy will play no role throughout our analysis*, thereby resolving issue (b).

**§ Langevin-Robbins-Monro Schemes.** We have seen that the LRM template in Section 2 is capable of capturing a broad range of existing and new algorithms in a unified way. To resolve the remaining issue (a), we will further rely on the LRM template: for proving that (LRM) generates a WAPT of the corresponding SDE, we show that the key condition (1) in WAPT can be reduced to checking an easy-to-verify bound on the perturbation terms $Z_k$.

To achieve this, the most important step in our proof, which distinguishes our analysis from all existing works in non-convex sampling, is the construction of the so-called *Picard process*, the natural generalization of the Picard's successive approximation method [17] from ordinary differential equations to *stochastic* differential equations. In the stochastic approximation literature, similar techniques have been successfully applied to study optimization and games in various settings such as on Riemannian or primal-dual spaces [26, 28, 37]. The application to sampling has also been previously explored by Bubeck et al. [11], Chau et al. [12] in different contexts. What distinguishes our work from the existing literature is the advantage of generalizing the Picard process to encompass a vastly wider class of algorithms, specifically the LRM schemes. Moreover, the integration of the Picard process with the theory of WAPT plays a pivotal role in our analysis, and both of these aspects present original contributions.

**§ Framework overview.** To conclude, for proving last-iterate convergence, we proceed as follows:

1. For a given LRM scheme $\{x_k\}_{k \in \mathbb{N}}$, we first construct a continuous-time trajectory $(X_t)_{t \geq 0}$ via *interpolating* the iterates (see (3)).

2. We prove that $(X_t)$ constitutes a WAPT of the SDE (see Theorem 1). This step relies heavily on the construction of the aforementioned Picard process.

3. By invoking the dynamical system theory of Benaïm and Hirsch [7], the convergence of LRM schemes reduces to simply checking the *stability condition* (Theorem 2). In the Wasserstein space, this condition translates into boundedness of the second moments of the iterates $\{x_k\}$, for which there is a plethora of approaches; we present two such methods in Section 5.

Fig. 1 depicts a high-level overview of the ingredients needed in our framework, and their corresponding theorems.

## 4 The Dynamics of Langevin-Robbins-Monro Schemes

In this section, we view (LRM) as a noisy and biased discretization of (LD). To make this analogy precise, let $(B_t)_{t \geq 0}$ be a Brownian motion defined on a filtered probability space with filtration

---

[3]Definition 1 is phrased in terms of a continuous-time stochastic process $(X_t)_{t \geq 0}$. The discrete iterates $\{x_k\}_{k \in \mathbb{N}}$ can be converted to a continuous-time process through a suitable interpolation; see (3).

$(\mathcal{F}_t)_{t\geq0}$, and define $\tau_k = \sum_{n=1}^{k} \gamma_n$ to be the effective time that has elapsed at the iteration $k$. Using the Brownian motion, we can rewrite (LRM) as

$$x_{k+1} = x_k - \gamma_{k+1}\{v(x_k) + Z_{k+1}\} + \sigma(x_k)(B_{\tau_{k+1}} - B_{\tau_k}), \tag{2}$$

assuming that the filtration satisfies $Z_k \in \mathcal{F}_{\tau_k}$.[4] The (continuous-time) *interpolation* $(X_t)_{t\geq0}$ of $\{x_k\}_{k\in\mathbb{N}}$ is then defined as the adapted process

$$X_t = x_k - (t - \tau_k)\{v(x_k) + \mathbb{E}[Z_{k+1} \mid \mathcal{F}_t]\} + \sigma(x_k)(B_t - B_{\tau_k}), \quad \text{for } t \in [\tau_k, \tau_{k+1}]. \tag{3}$$

In addition, for a fixed $t$, consider the Brownian motion $(B_s^{(t)})_{s\geq0}$ where $B_s^{(t)} := B_{t+s} - B_t$, and define the *Langevin flow* $(\Phi_s^{(t)})_{s\geq0}$ as the (strong) solution of (SDE) initialized at $X_t$. It is important to note that $\Phi^{(t)}$ and $X$ are synchronously coupled by sharing the same Brownian motion.

## 4.1 Technical Assumptions and Requirements

We now introduce the basic technical assumptions and discuss their generality.

**Assumption 1.** *The vector field $v$ is $L$-Lipschitz, and satisfies $\langle x, v(x) \rangle \leq C_v(1 + \|x\|)$ for some $C_v > 0$. Moreover, $\sigma$ is $L$-Lipschitz and is bounded in Hilbert-Schmidt norm.*

Lipschitzness of $v$ is a standard assumption and is also required to ensure the existence of a unique strong solution of (SDE). The second assumption on the vector field is exceedingly weak and when $v = -\nabla f$, is satisfied even for distributions without moments. The assumptions on diffusion coefficient $\sigma$ are already satisfied when $\sigma \equiv 1$, and we show that it holds for practical schemes such as Example 4.

**Assumption 2.** *The Robbins-Monro summability conditions hold: $\sum_{k=1}^{\infty} \gamma_k = \infty$ and $\sum_{k=1}^{\infty} \gamma_k^2 < \infty$. Moreover, for some constant $P$ to be defined in (20), we have*

$$\gamma_{k+1}/\gamma_k + P\gamma_k\gamma_{k+1} < 1 - \gamma_k, \quad \forall k. \tag{4}$$

The Robbins-Monro step size conditions are standard in the non-convex sampling literature [4, 20, 29, 30]. For (4), it can be verified that condition is satisfied even for slowly-decreasing step sizes such as $\gamma_k \propto (\sqrt{k} \log k)^{-1}$, which hence is not restrictive.

**Assumption 3.** *The noises $\{U_k\}_{k\in\mathbb{N}}$ form a martingale difference sequence, i.e., $\mathbb{E}[U_{k+1} \mid U_k] = 0$, and have uniformly bounded second moments. In addition, the bias terms satisfy*

$$\mathbb{E}[\|b_{k+1}\|^2 \mid \mathcal{F}_{\tau_k}] = O(\gamma_{k+1}^2\|v(x_k)\|^2 + \gamma_{k+1}). \tag{5}$$

A martingale difference sequence is more general than an i.i.d. sequence, allowing the noise to be state-dependent. The bias condition (5) simply states that the bias shall not overpower the signal $v(x_k)$, and, as we show later, is satisfied by all our examples.

## 4.2 From Discrete to Continuous: LRM Schemes and WAPTs

We are now in a position to state our main theorems. Our first result below establishes a precise link between the discrete-time (LRM) and the continuous-time (SDE).

**Theorem 1.** *Under Assumptions 1–3, the interpolation (3) of an LRM scheme is a Wasserstein asymptotic pseudotrajectory of (SDE).*

§ **Sketch of the Proof for Theorem 1.** The proof of this theorem is heavily based on the notion of the Picard process and iterate moment bounds. The complete proof can be found in Appendix C.

§ **Step 1: The Picard Process.** For a fixed $t > 0$, recall the construction of the interpolation (3) and the Langevin flow. Central to our analysis is the *Picard process*, defined as

$$Y_s^{(t)} = X_t + \int_0^s v(X_{t+u}) \, du + \int_0^s \sigma(X_{t+u}) \, dB_u^{(t)}. \tag{6}$$

---

[4]One can augment the filtration of the Brownian motion by the $\sigma$-algebra of $Z_k$ at times $\{\tau_k\}_{k\in\mathbb{N}}$.

The Picard process is adapted and is (synchronously) coupled with the Langevin flow and the interpolation. We think of the Picard process as one step of the *Picard iteration* for successive approximations to solve ODEs. This means, intuitively, that its trajectory should be close to the original interpolation, as well as to that of the Langevin flow, playing the role of a "bridge".

Fix $T > 0$. For $s \in [0, T]$, we decompose the distance between the interpolation $X_t$ in (3) and the Langevin flow as

$$\frac{1}{2}\|X_{t+s} - \Phi_s^{(t)}\|^2 \leq \|Y_s^{(t)} - \Phi_s^{(t)}\|^2 + \|X_{t+s} - Y_s^{(t)}\|^2. \tag{7}$$

We now bound each term of the decomposition. By synchronous coupling of the processes, Lipschitzness of $v$, and Itô isometry, Lemma 3 bounds the first term as

$$\|Y_s^{(t)} - \Phi_s^{(t)}\|^2 \leq 2(T+1)L^2 \int_0^s \|\Phi_u^{(t)} - X_{t+u}\|^2 \, du. \tag{8}$$

This will be suitable for later use of Grönwall's lemma.

**§ Step 2: Accumulated Noise and Bias.** For the rest of the proof, we need some extra notation. Define $m(t) := \sup\{k \geq 0 : \tau_k \leq t\}$ and the piecewise-constant process $\overline{X}_t := x_{m(t)}$. Going back to the second term of (7), observe that

$$X_{t+s} - Y_s^{(t)} = \int_t^{t+s} v(\overline{X}_u) - v(X_u) \, du + \int_0^s \sigma(\overline{X}_{t+u}) - \sigma(X_{t+u}) \, dB_u^{(t)} - \Delta_Z(t, s), \tag{9}$$

where $\Delta_Z(t, s)$ is the accumulated noise and bias from time $t$ to time $t + s$. It is expected that $\|\Delta_Z(t, s)\|$ eventually becomes negligible, since the step size becomes small. The next lemma confirms this intuition.

**Lemma 1.** *Suppose Assumptions 1–3 hold. Then, for any fixed $T > 0$ we have*

$$\lim_{t \to \infty} \sup_{0 \leq s \leq T} \mathbb{E}\|\Delta_Z(t, s)\|^2 = 0.$$

**§ Step 3: Gradient Moment Bounds.** Based on (9) and Lemma 1, bounding the distance between the Picard process and the interpolation essentially reduces to bounding how much the discrete algorithm "moves" during one iteration in expectation. This, in turn, depends on how large the moments of $\|v(x_k)\|$ grow per iteration, which is controlled by the following lemma:

**Lemma 2.** *Let $\{x_k\}_{k \in \mathbb{N}}$ be the iterates of (LRM) and suppose Assumptions 1–3 hold. Then, $\mathbb{E}\|x_k\|^2 = O(1/\gamma_{k+1})$. This in turn implies $\mathbb{E}\|v(x_k)\|^2 = O(1/\gamma_{k+1})$ and $\mathbb{E}\|b_{k+1}\|^2 = O(\gamma_{k+1})$.*

Using this lemma and Lemma 1 we can obtain $A_t := \sup_{0 \leq s \leq T} \mathbb{E}\|X_{t+s} - Y_s^{(t)}\|^2 \to 0$ as $t \to \infty$, which shows that the Picard process gets arbitrarily close to the interpolation as $t \to \infty$.

**§ Step 4: Concluding the Proof.** Let us go back to the decomposition (7). Taking expectation and using (8) and Grönwall's lemma, we obtain $\mathbb{E}[\|X_{t+s} - \Phi_s^{(t)}\|^2] \leq 4 A_t \exp(T^2 L^2)$, Thus,

$$\lim_{t \to \infty} \sup_{s \in [0, T]} \mathbb{E}\left[\|X_{t+s} - \Phi_s^{(t)}\|^2\right] = 0.$$

As we coupled $X_{t+s}$ and $\Phi_s^{(t)}$ in a specific way (via synchronizing the Brownian motions), we directly get an upper bound on the Wasserstein distance. ∎

## 5 Last-Iterate Convergence of Sampling Schemes

In this section we focus on last-iterate convergence of LRM schemes in Wasserstein space. We first explore the interplay between the convergence of WAPTs and *stability*. We then show that the existing stability results for simple Euler-Maruyama discretization of the Langevin diffusion can be extended, with little to no extra assumptions, to the class of LRM schemes in Section 2. This in turn readily implies the last-iterate convergence of a wide class of LRM schemes.

## 5.1 From WAPTs to Convergence in $\mathbb{W}_2$

Since convergence of the distribution of $x_k$ to $\pi$ in Wasserstein distance implies convergence of the second moments of $x_k$ to that of $\pi$ [3], convergence in the Wasserstein space should at least require:

$$\sup_{k \in \mathbb{N}} \mathbb{E}\|x_k\|^2 < \infty. \tag{10}$$

It turns out that, for WAPTs, the exceedingly weak necessary condition (10) is also *sufficient*:

**Theorem 2.** *Let $(X_t)$ be a Wasserstein asymptotic pseudotrajectory of the Langevin diffusion* (LD) *generated by an LRM scheme $\{x_k\}$ via* (3). *Then $W_2(x_k, \pi) \to 0$ if and only if* (10) *holds.*

*Proof.* The proof relies on the structure of compact sets in the Wasserstein space and limit-set theorems for dynamical systems [7]. Specifically, the closure of bounded subsets of $\mathbb{W}_2$ is compact [3], so condition (10) implies that $(\text{law}(X_t))_{t \geq 0}$ is pre-compact in $\mathbb{W}_2$. Moreover, Assumption 1 implies that the Langevin flow is globally integrable. Thus, $(\text{law}(X_t))_{t \geq 0}$ is a pre-compact WAPT of a globally integrable flow, and we can apply the limit-set theorem for metric spaces [7, Theorem 0.1] to conclude that the limit-set of $(\text{law}(X_t))_t$ is an *internally chain transitive (ICT) set*.

Next, we show that for the case of the Langevin flow, the only ICT set is $\{\pi\}$, implying the desired convergence of our theorem. To see this, define $V(\cdot) = D_{\text{KL}}(\cdot \mid \pi)$. It can be observed that $V$ is a Lyapunov function for (LD), whose value is strictly decreasing along the flow (as the time derivative of $V$ along the flow is negative of the relative Fisher information, which is strictly positive for all measures other than $\pi$). Thus, all requirements of [6, Prop. 6.4] are satisfied, showing that the only point in the ICT set is $\pi$. This also shows the uniqueness of the stationary distribution of (LD). ∎

*Remark.* From the proof of Theorem 1, we observe that the supremum of the Wasserstein distance between $(X_.)_{[t,t+T]}$ and $(\Phi_.^{(t)})_{[0,T]}$ typically scales exponentially with $T$, which is common for weak approximation error in the literature, see [40]. Despite the exponential dependence on $T$, the convergence of the last iterate is assured by Theorem 2 without a need of a uniform control in $T$. This is primarily attributed to the adoption of a dynamical system viewpoint and the application of corresponding tools, effectively harnessing the paradigm established by Benaïm and Hirsch.

Theorems 1–2 in tandem thus show that, as long as an LRM scheme satisfies Assumptions 1–3 and the moment condition (10), the desirable last-iterate convergence in $\mathbb{W}_2$ is immediately attained. Therefore, in the rest of this section, we turn our focus to establishing (10) for LRM schemes.

## 5.2 Bounded Moments of LRM Schemes

There is a long history of study on conditions that ensure (10) for iterative algorithms, which has culminated in the so-called *dissipativity* properties. We consider two such examples below.

**Assumption 4** (Dissipativity)**.** *There exist constants $\alpha > 0$ and $\beta \geq 0$ such that*

$$\langle x, v(x) \rangle \leq -\alpha\|x\|^2 + \beta, \quad \forall x \in \mathbb{R}^d.$$

Under Assumption 4, it is classical that (10) holds for the simple Euler-Maruyama discretization of (LD) with deterministic or stochastic gradient oracles [23, 29, 30, 38, 48, 51, 54]. These studies, however, cannot handle *non-zero bias*, which, as seen in Examples 1–3, is crucial for incorporating more advanced sampling schemes.

To this end, our next result shows that for a wide class of LRM schemes, the stability (10) essentially comes for free under Assumption 4. The proof is provided in Appendix D.

**Theorem 3.** *Let $v$ be a vector field satisfying Assumptions 1 and 4 and $\sigma$ be a diffusion coefficient satisfying Assumption 1, and let $\{x_k\}$ be an LRM scheme. Assume that $\lim_{k \to \infty} \gamma_k = 0$, $\sup_k \mathbb{E}\|U_k\|^2 < \infty$, and the bias satisfies* (5). *Then, the stability condition* (10) *holds for $\{x_k\}$.*

A weaker notion of dissipativity that has been studied in the literature is:

**Assumption 5** (Weak dissipativity)**.** *There exist constants $\alpha > 0$, $\kappa \in (0, 1]$, and $\beta \geq 0$ such that*

$$\langle x, v(x) \rangle \leq -\alpha\|x\|^{1+\kappa} + \beta, \quad \forall x \in \mathbb{R}^d.$$

| | NOISE | BIAS | LAST-ITERATE |
|---|---|---|---|
| LAMBERTON AND PAGES [29], LEMAIRE [30] | ✗ | ✗ | ✗ |
| TEH ET AL. [54] | ✓ | ✗ | ✗ |
| BENAÏM ET AL. [8] | ✗ | ✗ | ✓ |
| DURMUS AND MOULINES [20] | ✗ | ✗ | ✓ |
| BALASUBRAMANIAN ET AL. [4] | ✗ | ✗ | ✗ |
| THIS WORK | ✓ | ✓ | ✓ |

Table 1: Comparison to existing works on convergence of LRM schemes. All methods, except for [4], require bounded second moments of the iterates.

When $\kappa = 1$, Assumption 5 is simply Assumption 4. As opposed to Assumption 4, which requires *quadratic growth* of $f$ outside a compact set (when $v = -\nabla f$), Assumption 5 only entails *superlinear growth* and therefore is considerably weaker.

For Euler-Maruyama discretization of (LD) with deterministic gradients, [20] prove that Assumption 5 is sufficient to guarantee bounded moments of the iterates. As for a generic LRM scheme, we consider the following general condition on the bias terms, which will suffice to cover all our examples in Section 2: For some constant $c$,

$$\|b_{k+1}\|^2 \leq c\big(\gamma_{k+1}^2\|v(x_k)\|^2 + \gamma_{k+1}^2\|U'_{k+1}\|^2 + \gamma_{k+1}\|\xi'_{k+1}\|^2 + \gamma_{k+1}\|\xi_{k+1}\|^2\big), \tag{11}$$

where $U'_{k+1}$ is an extra noise term, and $\xi'_{k+1}$ is a standard Gaussian independent of the noises and $\xi_k$. The price to pay with the weaker Assumption 5, however, is that we need to assume sub-Gaussianity of the noise. For a proof, see Appendix D.

**Theorem 4.** *Let $\pi \propto e^{-f}$ be the target distribution, where $v = -\nabla f$ satisfies Assumptions 1 and 5, and let $\{x_k\}$ be an LRM scheme. Assume that $\lim_{n\to\infty} \gamma_k = 0$, the noises $U_k$ and $U'_k$ are sub-Gaussian, and the bias term of $\{x_k\}$ satisfies (11). Then, (10) holds for $\{x_k\}$ in when (i) $\sigma \equiv 1$, or (ii) $f$ is Lipschitz and the LRM follows the Mirror Langevin algorithm (Example 4).*

### 5.3 Examples of Convergent LRM Schemes

We now illustrate the use of Theorems 1–4 on our examples in Section 2.

**Proposition 1.** *Under Assumption 1 and noise with uniformly bounded second moments, the following holds for Examples 1–6: (i) The bias has the form (11) and satisfies (5), (ii) As a result, under Assumptions 2 and 3, Examples 1–6 produce iterates that generate a WAPT of (SDE). (iii) Under the additional conditions of Theorem 3 or Theorem 4, Examples 1–6 enjoy last-iterate convergence to the target distribution in Wasserstein distance.*

### 5.4 Comparison to Existing Work

We now give a more detailed comparison of our results to existing literature; a summary is given in Table 1, and additional comparison with prior works can be found in Appendix B.

**§ Guarantees for LRM Schemes.** Lamberton and Pages [29] and Lemaire [30] study the simple Euler-Maruyama discretization of (LD) with deterministic gradients (i.e., $U_k = b_k = 0$) and establish the weak convergence of the average iterates under a moment condition that is slightly weaker than (10).[5] Their analysis is further extended by [54] to incorporate stochastic gradients. Later, the last-iterate convergence of the simple Euler-Maruyama discretization of (LD) is studied by [20], who prove the convergence in the total variation distance under Assumption 5. Another work on a similar setting as [20] is [8], where the convergence criterion is given in an integral probability metric (IPM) [42] of the form $d_{\mathcal{B}}(\mu, \nu) := \sup_{\varphi \in \mathcal{B}} |\mathbb{E}_\mu \varphi - \mathbb{E}_\nu \varphi|$ for a certain class of test functions $\mathcal{B}$ that is known to imply weak convergence, but not convergence in total variation or Wasserstein distances.

Compared to these results, our guarantees possess the following desirable features:

---

[5] Although the condition in [29, 30] is stated in a weaker form than (10), it is typically only verified on a special case that is equivalent to our Assumption 4, and thus implies (10). See e.g., [29, Remark 3].

- The convergence is always on the last iterates instead of the average iterates.

- As we tolerate biased algorithms, the class of LRM schemes we consider is significantly more general than the ones in existing work.

Finally, we note that our results are incomparable to the recent work of Balasubramanian et al. [4], who derive the same result as in [29, 30], i.e., average-iterate, weak convergence, deterministic Euler-Maruyama discretization. A remarkable feature of the analysis in [4] is that it does not require any bounded moments, and, in particular, their bounds can be applied to target distributions with unbounded variance. However, the downside of [4] is that, in the presence of $U_k$ and $b_k$, their analysis produces a bound that does *not* vanish as $k \to \infty$; see [4, Theorem 15]. In contrast, our framework can tolerate quite general $U_k$ and $b_k$, gives stronger guarantees ($W_2$ vs. weak convergence; last-iterate vs. average-iterate).

**§ On Analysis Techniques.**   While, to our knowledge, our framework is significantly different from previous works on sampling, we acknowledge that similar ideas of creating an auxiliary process in-between the iterates and the continuous-time flow is not entirely new and has been touched upon in the literature, e.g., [10, 12]. That being said, our specific approach in building the Picard process and its development into a wider array of algorithms, i.e., Langevin-Robbins-Monro schemes, undoubtedly plays a pivotal role in our analysis. Moreover, the integration of the Picard process with the theory of asymptotic pseudo-trajectories offers dual benefits to our study, and we view these as our unique contributions to this area of research.

Furthermore, the novel Picard process gives a significant advantage in all of our results. The work of [8] also hinges on dynamical system theory-related ideas. Yet, missing the critical step of the Picard process has seemingly resulted in much weaker findings compared to our work. This observation is not meant as a critique; rather, it merely highlights the potency of the unique method we have integrated into our study.

## 6   Concluding Remarks

In this paper, we provided a new, unified framework for analyzing a wide range of sampling schemes, thus laying the theoretical ground for using them in practice, as well as motivating new and more efficient sampling algorithms that enjoy rigorous guarantees. We built on the ideas from dynamical system theory, and gave a rather complete picture of the asymptotic behavior of many first-order sampling algorithms. In short, our results help with the following:

- **Validating existing methods:** Methods like mirror Langevin and randomized mid-point currently lack even asymptotic guarantees in fully non-convex scenarios, such as sampling from neural network-defined distributions. Our work fills this gap by offering the first rigorous justification for these schemes, supporting practitioners in utilizing these methods confidently.

- **Facilitating new algorithm design:** Our work motivates novel sampling methods through a straightforward verification of Assumptions 1–3. An illustrative instance involves the randomized mid-point method and Runge-Kutta integrators, wherein a substantial 50% reduction in computation per iteration can be achieved without compromising convergence by simply recycling past gradients, shown in Example 2. The balance between the benefits of saving gradient oracles and potential drawbacks remains an open question, necessitating case-by-case practical evaluation. Nevertheless, our theory provides a flexible algorithmic design template that extends beyond the current literature's scope.

While our WAPT result holds under very mild conditions, a severe limitation of our current framework is that it *only* applies to Langevin-based algorithms, whereas there exist numerous practical sampling schemes, such as Metropolis-Hastings, that are not immediately linked to (LD). We believe that this restriction arises as an artifact of our analysis, as the WAPT framework can in principle be applied equally well to any continuous-time dynamics. Lifting such constraint is an interesting future work.

## Acknowledgments and Disclosure of Funding

This work was supported by the European Research Council (ERC) under the European Union's Horizon 2020 research and innovation program grant agreement No 815943. YPH acknowledges funding through an ETH Foundations of Data Science (ETH-FDS) postdoctoral fellowship.

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

## A    Further Examples of LRM Schemes

**Example 5.** The classic *Stochastic Gradient Langevin Dynamics* [57] iterates as

$$x_{k+1} = x_k - \gamma_{k+1}\widetilde{\nabla}f(x_k) + \sqrt{2\gamma_{k+1}}\,\xi_{k+1}, \qquad\qquad \text{(SGLD)}$$

where $\widetilde{\nabla}f$ is the gradient of the negative log-likelihood of a random batch of the data. (SGLD) fits the LRM template by setting $U_{k+1} := \widetilde{\nabla}f(x_k) - \nabla f(x_k)$, and $b_{k+1} := 0$. □

**Example 6.** The *Proximal Langevin Algorithm* [9, 46, 59] is defined via

$$x_{k+1} = x_k - \gamma_{k+1}\nabla f(x_{k+1}) + \sqrt{2\gamma_{k+1}}\,\xi_{k+1}. \qquad\qquad \text{(PLA)}$$

This algorithm is implicit, and it is assumed that one can solve (PLA) for $x_{k+1}$. By setting $b_{k+1} := \nabla f(x_{k+1}) - \nabla f(x_k)$ and $U_{k+1} := 0$, we see that this algorithm also follows the LRM template. □

## B    Additional Related Work

Our paper studies the behavior of a wide range of Langevin-based sampling algorithms proposed in the literature in the asymptotic setting under minimal assumptions. This allows us to give last-iterate guarantees in Wasserstein distance. As stressed in Section 1, our goal is *not* to provide non-asymptotic rates in this general setting as the problem is inherently NP-Hard. However, given more assumptions and structures on the potential $f$, there is a plethora of works which prove convergence rates for the last iterates in Wasserstein distance. In this appendix, we provide additional backgraound for these works and the methods used in the literature.

A powerful framework for quantifying the global discretization error of a numerical algorithm is the mean-square analysis framework [40]. This framework furnishes a general recipe for controlling short and long-term integration errors. For sampling, this framework has been applied to prove convergence rates for Langevin Monte-Carlo (the Euler-Maruyama discretization of (LD)) in the strongly-convex setting [32, 34]. Similar to our work, the convergence obtained in these works is last-iterate and in Wasserstein distance. One of the essential ingredients in the latter work is the contraction property of the SDE, which is ensured by the strong convexity assumption. This, in turn, implies strong non-asymptotic convergence guarantees.

It is an interesting future direction to study the combination of the Mean-Squared analysis together with the Picard process and its applicability to more sophisticated algorithms (such as LRM schemes with bias and noise), as well as non-convex potentials.

As explained in Section 3, one of the main themes in proving error bounds for sampling is the natural relation between sampling and optimization in the Wasserstein space. This point of view, when applied to strongly-convex potentials, has produced numerous non-asymptotic guarantees; see [14, 18] for a recent account and the references therein. Note that strong convexity is crucial for the analysis used in the aforementioned work. Moreover, the error bounds for biased and noisy discretizations do *not* decrease with the step-size or iteration count; see [18, Theorem 4, Eqn. (14)]. This means that while the bound is non-asymptotic, it does not automatically result in an asymptotic convergence. Finally, we stress that these approaches are orthogonal to our techniques: We view a sampling algorithm as a (noisy and biased) discretization of a dynamical system (and not necessarily a gradient flow), and use tools from dynamical system theory to provide asymptotic convergence results.

## C    Proofs for Section 4

### C.1    Proof of Theorem 1

In this appendix, we bring the detailed proof of Theorem 1. Recall that we interpolate the iterates of the LRM scheme $\{x_k\}$ as

$$X_t = x_k + (t - \tau_k)\{v(x_k) + \mathbb{E}[Z_{k+1} \mid \mathscr{F}_t]\} + \sigma(x_k)(B_t - B_{\tau_k}). \qquad\qquad \text{(3)}$$

Moreover, for a fixed $t > 0$, we considered the Brownian motion $B_s^{(t)} = B_{t+s} - B_t$, and constructed two important processes: the Langevin flow defined via

$$d\Phi_s^{(t)} = v(\Phi_s^{(t)})\,ds + \sigma(\Phi_s^{(t)})\,dB_s^{(t)}, \quad \Phi_0^{(t)} = X_t, \qquad\qquad \text{(12)}$$

and the Picard process (6) constructed as

$$Y_s^{(t)} = X_t + \int_0^s v(X_{t+u}) \, du + \int_0^s \sigma(X_{t+u}) \, dB_u^{(t)}. \tag{6}$$

Let us fix $T > 0$, and for $s \in [0, T]$ decompose the distance between the interpolation and the Langevin flow as

$$\tfrac{1}{2}\|X_{t+s} - \Phi_s^{(t)}\|^2 \leq \|Y_s^{(t)} - \Phi_s^{(t)}\|^2 + \|X_{t+s} - Y_s^{(t)}\|^2, \tag{7}$$

where we have used $\|a + b\|^2 \leq 2\|a\|^2 + 2\|b\|^2$. We now bound each term of this decomposition. Notice that due to the synchronous coupling of the processes, the Brownian motion cancels out in the differences.

The first term controls how close the Picard process is to the Langevin flow, and is bounded in the following lemma.

**Lemma 3.** *For fixed $t, T > 0$ and $0 \leq s \leq T$, the distance of the Picard process and the Langevin flow is bounded as*

$$\|Y_s^{(t)} - \Phi_s^{(t)}\|^2 \leq 2(T + 1)L^2 \int_0^s \|\Phi_u^{(t)} - X_{t+u}\|^2 \, du.$$

*Proof.* By the auxiliary Lemma 4 below, Lipschitzness of $v, \sigma$, Itô isometry (see, e.g., [62]) and $s \leq T$, we have

$$\mathbb{E}\|Y_s^{(t)} - \Phi_s^{(t)}\|^2 = \mathbb{E}\left\|\int_0^s v(\Phi_u^{(t)}) - v(X_{t+u}) \, du + \int_0^s \sigma(\Phi_u^{(t)}) - \sigma(X_{t+u}) \, dB_u^{(t)}\right\|^2$$

$$\leq 2s \int_0^s \mathbb{E}\left\|v(\Phi_u^{(t)}) - v(X_{t+u})\right\|^2 \, du + 2\mathbb{E}\int_0^s \left\|\sigma(X_{t+u}) - \sigma(\Phi_u^{(t)})\right\|_F^2 \, du$$

$$\leq 2(T + 1)L^2 \int_0^s \mathbb{E}\|\Phi_u^{(t)} - X_{t+u}\|^2 \, du. \qquad \blacksquare$$

For the rest of the proof, we need to define the continuous-time piecewise-constant processes $\overline{X}(\tau_k + s) = X_k$, $\overline{\gamma}(\tau_k + s) = \gamma_{k+1}$, $\overline{Z}(\tau_k + s) = Z_{k+1}$, and $Z(\tau_k + s) = \mathbb{E}[Z_{k+1} \,|\, \mathscr{F}_{\tau_k+s}]$, for $0 \leq s < \gamma_{k+1}$. Also, let $m(t) = \sup\{k \geq 0 : \tau_k \leq t\}$ so that $\tau_{m(t)} \leq t < \tau_{m(t)+1}$.

To bound the second term in (7), we have seen that

$$X_{t+s} - Y_s^{(t)} = \int_t^{t+s} v(\overline{X}(u)) \, du - \int_0^s v(X_{t+u}) \, du$$

$$+ \int_t^{t+s} \sigma(\overline{X}(u)) \, dB_u - \int_0^s \sigma(X_{t+u}) \, dB_u^{(t)}$$

$$+ \Delta_Z(t, s),$$

where $\Delta_Z(t, s)$ plays the role of accumulated noise and bias from time $t$ to $t + s$, and is defined as

$$\Delta_Z(t, s) := \sum_{i=n}^{k-1} \gamma_{i+1} Z_{i+1} + (t + s - \tau_k)\mathbb{E}[Z_{k+1} \,|\, \mathscr{F}_{t+s}] - (t - \tau_n)\mathbb{E}[Z_{n+1} \,|\, \mathscr{F}_t], \tag{13}$$

with $k = m(t + s)$ and $n = m(t)$. We therefore have

$$\mathbb{E}\|X_{t+s} - Y_s^{(t)}\|^2 \leq 3\mathbb{E}\left\|\int_t^{t+s} v(X_u) - v(\overline{X}(u)) \, du\right\|^2$$

$$+ 3\mathbb{E}\left\|\int_t^{t+s} \sigma(X_u) - \sigma(\overline{X}(u)) \, dB_u\right\|^2 + 3\mathbb{E}\|\Delta_Z(t, s)\|^2$$

$$\leq 3s \int_t^{t+s} \mathbb{E}\left\|v(X_u) - v(\overline{X}(u))\right\|^2 \, du$$

$$+ 3\mathbb{E}\int_t^{t+s} \left\|\sigma(X_u) - \sigma(\overline{X}(u))\right\|_F^2 \, du + 3\mathbb{E}\|\Delta_Z(t, s)\|^2$$

$$\leq 3(s + 1)L^2 \int_t^{t+s} \mathbb{E}\|X_u - \overline{X}(u)\|^2 \, du + 3\mathbb{E}\|\Delta_Z(t, s)\|^2. \tag{14}$$

For bounding the term inside the integral, we have

$$\mathbb{E}\|X_u - \overline{X}(u)\|^2 = \mathbb{E}\|(u - \tau_{m(u)})\{v(\overline{X}(u)) + Z(u)\} + \sigma(\overline{X}(u))\,(B_u - B_{\tau_{m(u)}})\|^2$$

$$\leq 4\overline{\gamma}(u)^2\Big(\mathbb{E}\|v(\overline{X}(u))\|^2 + \mathbb{E}\|Z(u)\|^2\Big) + 2\overline{\gamma}(u)\,\mathbb{E}\,\mathrm{tr}\Big(\sigma(\overline{X}(u))^\top\sigma(\overline{X}(u))\Big).$$

We have used the fact that

$$\mathbb{E}\|\sigma(\overline{X}(u))\,(B_u - B_{\tau_{m(u)}})\|^2 = \mathbb{E}\Big((B_u - B_{\tau_{m(u)}})^\top\sigma(\overline{X}(u))^\top\sigma(\overline{X}(u))\,(B_u - B_{\tau_{m(u)}})\Big)$$

$$= \mathbb{E}\mathrm{tr}\Big(\sigma(\overline{X}(u))^\top\sigma(\overline{X}(u))\,(B_u - B_{\tau_{m(u)}})(B_u - B_{\tau_{m(u)}})^\top\Big)$$

$$= \mathbb{E}\Big[\mathbb{E}[\mathrm{tr}(\sigma(\overline{X}(u))^\top\sigma(\overline{X}(u))\,(B_u - B_{\tau_{m(u)}})(B_u - B_{\tau_{m(u)}})^\top)\mid\mathscr{F}_{\tau_{m(u)}}]\Big]$$

$$= (u - \tau_{m(u)})\mathbb{E}\Big[\mathrm{tr}(\sigma(\overline{X}(u))^\top\sigma(\overline{X}(u)))\Big]$$

Notice that since conditional expectation is a projection in $L^2$, we have $\mathbb{E}\|Z(u)\|^2 \leq \mathbb{E}\|\overline{Z}(u)\|^2$. Using this fact, along with boundedness of $\sigma(\cdot)$ by $C_\sigma$, and [Lemma 2](#) we get

$$\mathbb{E}\Big[\|X_u - \overline{X}(u)\|^2\Big] \leq 4\overline{\gamma}(u)^2\Big(\mathbb{E}\|v(\overline{X}(u))\|^2 + \mathbb{E}\|\overline{Z}(u)\|^2\Big) + 2\overline{\gamma}(u)\,\mathbb{E}\,\mathrm{tr}\Big(\sigma(\overline{X}(u))^\top\sigma(\overline{X}(u))\Big)$$

$$\leq 4\overline{\gamma}(u)^2\mathbb{E}\|v(\overline{X}(u))\|^2 + 8\overline{\gamma}(u)^2\sigma^2 + 4\overline{\gamma}(u)^2\,O(\overline{\gamma}(u)) + 2C_\sigma\overline{\gamma}(u) \leq C\overline{\gamma}(u),$$

for some constant $C > 0$. Plugging this estimate into [(14)](#) after taking expectation yields

$$\mathbb{E}\Big[\|X_{t+s} - Y_s^{(t)}\|^2\Big] \leq 3(s+1)L^2C\int_t^{t+s}\overline{\gamma}(u)\,\mathrm{d}u + 3\mathbb{E}\|\Delta_Z(t,s)\|^2$$

$$\leq 3(s+1)sL^2C\sup_{u\in[t,t+s]}\overline{\gamma}(u) + 3\mathbb{E}\|\Delta_Z(t,s)\|^2$$

$$\leq 3(T+1)^2L^2C\sup_{u\in[t,t+T]}\overline{\gamma}(u) + 3\sup_{u\in[0,T]}\mathbb{E}\|\Delta_Z(t,u)\|^2$$

Taking supremum over $s \in [0,T]$ and noticing that the right-hand-side is independent of $s$ and $\gamma_k \to 0$, together with [Lemma 1](#) yields

$$A_t := \sup_{0\leq s\leq T}\mathbb{E}\Big[\|X_{t+s} - Y_s^{(t)}\|^2\Big] \tag{15}$$

$$\leq 3(T+1)^2L^2C\sup_{t\leq u\leq t+T}\overline{\gamma}(u) + 3\sup_{0\leq u\leq T}\mathbb{E}\Big[\|\Delta_Z(t,u)\|^2\Big]$$

$$\to 0 \quad \text{as } t \to \infty,$$

showing that the Picard process gets arbitrary close to the original interpolation, as $t \to \infty$.

Let us return to the decomposition [(7)](#). By taking expectation and using [(8)](#) and [(15)](#) we obtain

$$\mathbb{E}\Big[\|X_{t+s} - \Phi_s^{(t)}\|^2\Big] \leq 2(T+1)L^2\int_0^s\mathbb{E}\Big[\|X_{t+u} - \Phi_u^{(t)}\|^2\Big]\,\mathrm{d}u + 2A_t$$

$$\leq 2A_t\exp\Big(s(T+1)L^2\Big)$$

$$\leq 2A_t\exp((T+1)^2L^2),$$

where in the last line we have used the Grönwall lemma. Thus,

$$\lim_{t\to\infty}\sup_{s\in[0,T]}\mathbb{E}\Big[\|X_{t+s} - \Phi_s^{(t)}\|^2\Big] = 0.$$

Recall that the Wasserstein distance between $X_{t+s}$ and $\Phi_s^{(t)}$ is the infimum over all possible couplings between them, having the correct marginals. As $\Phi_s^{(t)}$ has the same marginal as the Langevin diffusion started from $X_t$ at time $s$, and the synchronous coupling of the interpolation and the Langevin flow produces a specific coupling between them, we directly get

$$W_2(X_{t+s}, \Phi_s^{(t)}) \leq \mathbb{E}\Big[\|X_{t+s} - \Phi_s^{(t)}\|^2\Big]^{\frac{1}{2}},$$

which implies

$$\lim_{t\to\infty}\sup_{s\in[0,T]}W_2(X_{t+s}, \Phi_s^{(t)}) = 0,$$

as desired. ∎

## C.2 Auxiliary Lemmas

**Lemma 1.** *Suppose Assumptions 1–3 hold. Then, for any fixed $T > 0$ we have*

$$\lim_{t\to\infty} \sup_{0\le s\le T} \mathbb{E}\|\Delta_Z(t,s)\|^2 = 0.$$

*Proof.* Define $\Delta_b$ and $\Delta_U$ the same way as in (13). By Cauchy-Schwarz we have

$$\|\Delta_b(t,s)\|^2$$

$$\le \left(\sum_{i=n}^{k-1} \gamma_{i+1}\|b_{i+1}\| + (t+s-\tau_k)\|\mathbb{E}[b_{k+1}\mid\mathscr{F}_{t+s}]\| + (t-\tau_n)\|\mathbb{E}[b_{n+1}\mid\mathscr{F}_t]\|\right)^2$$

$$\le (2\gamma_{n+1}+s)\left(\sum_{i=n}^{k-1} \gamma_{i+1}\|b_{i+1}\|^2 + (t+s-\tau_k)\|\mathbb{E}[b_{k+1}\mid\mathscr{F}_{t+s}]\|^2 + (t-\tau_n)\|\mathbb{E}[b_{n+1}\mid\mathscr{F}_t]\|^2\right),$$

where the last inequality comes from $\sum_{i=n}^{k-1}\gamma_{i+1} \le s, t+s-\tau_k \le \gamma_{k+1}, t-\tau_n \le \gamma_{n+1}$, and $\gamma_{k+1} \le \gamma_{n+1}$.

Noticing that conditional expectation is a contraction in $L^2$ and letting $k' = m(t+T)$, we get

$$\sup_{0\le s\le T} \mathbb{E}\left[\|\Delta_b(t,s)\|^2\right] \le (2+T)\left(\sum_{i=n}^{k'-1} \gamma_{i+1}\mathbb{E}\|b_{i+1}\|^2 + \sup_{n\le j\le k'+1} \gamma_{j+1}\mathbb{E}\|b_{j+1}\|^2 + \gamma_{n+1}\mathbb{E}\|b_{n+1}\|^2\right)$$

Now, invoking Lemma 2 yields

$$\sup_{0\le s\le T} \mathbb{E}\left[\|\Delta_b(t,s)\|^2\right] \le C(2+T)\left(\sum_{i=n}^{k'-1} \gamma_{i+1}^2 + \sup_{n\le j\le k'+1} \gamma_{j+1}^2 + \gamma_{n+1}^2\right)$$

$$\le C(2+T)\left(\sum_{i=n}^{k'-1} \gamma_{i+1}^2 + 2\gamma_{n+1}^2\right)$$

$$\le C(2+T)(T+2\gamma_{n+1}) \sup_{0\le s\le T} \overline{\gamma}(t+s).$$

As $t \to \infty$, the last quantity vanishes, since $\gamma_n \to 0$.

For the noise we have

$$\|\Delta_U(t,s)\|^2 \le 2\left\|\sum_{i=n}^{k-1} \gamma_{i+1}U_{i+1}\right\|^2 + 4\|(t+s-\tau_k)\mathbb{E}[U_{k+1}\mid\mathscr{F}_{t+s}]\|^2 + 4\|(t-\tau_n)\mathbb{E}[U_{n+1}\mid\mathscr{F}_t]\|^2$$

$$\le 2\left\|\sum_{i=n}^{k-1} \gamma_{i+1}U_{i+1}\right\|^2 + 4\gamma_{k+1}^2\|U_{k+1}\|^2 + 4\gamma_{n+1}^2\|U_{n+1}\|^2.$$

Taking expectations and then sup, we get

$$\sup_{0\le s\le T} \mathbb{E}\left[\|\Delta_U(t,s)\|^2\right] \le 2 \sup_{n+1\le k\le m(t+T)} \mathbb{E}\left\|\sum_{i=n}^{k-1} \gamma_{i+1}U_{i+1}\right\|^2 + 4\gamma_{k+1}^2\sigma^2 + 4\gamma_{n+1}^2\sigma^2.$$

Since $\{U_i\}$ is a martingale difference sequence, we have that $\left\{\sum_{i=n}^{k-1}\gamma_{i+1}U_{i+1}\right\}_{k>n}$ is a martingale. Thus, by the boundedness of the second moments of $U_i$, we get

$$\mathbb{E}\left\|\sum_{i=n}^{k-1} \gamma_{i+1}U_{i+1}\right\|^2 = \sum_{i=n}^{k-1} \gamma_{i+1}^2\mathbb{E}\|U_{i+1}\|^2 \le \sigma^2 \sum_{i=n}^{k-1} \gamma_{i+1}^2.$$

Hence,

$$\lim_{n\to\infty} \sup\left\{\mathbb{E}\|\sum_{i=n}^{k-1} \gamma_{i+1}U_{i+1}\|^2 : n < k \le m(\tau_n+T)\right\} \le \lim_{n\to\infty} \sigma^2 \sum_{i=n}^{\infty} \gamma_{i+1}^2 = 0.$$

∎

**Lemma 2.** *Let $\{x_k\}_{k\in\mathbb{N}}$ be the iterates of* (LRM) *and suppose Assumptions 1–3 hold. Then, $\mathbb{E}\|x_k\|^2 = O(1/\gamma_{k+1})$. This in turn implies $\mathbb{E}\|v(x_k)\|^2 = O(1/\gamma_{k+1})$ and $\mathbb{E}\|b_{k+1}\|^2 = O(\gamma_{k+1})$.*

*Proof.* Without loss of generality, suppose $v$ has a stationary point at 0. We repeatedly use the fact that $\mathbb{E}\|v(x_k)\|^2 \le L^2\mathbb{E}\|x_k\|^2$. Moreover, by Assumption 1 we have $\langle v(x), x\rangle \le C_v(\|x\|+1)$, and $\|\sigma(x)\|_F^2 \le C_\sigma$.

Define $a_k := \mathbb{E}\|x_k\|^2$. We have

$$
\begin{aligned}
a_{k+1} - a_k &= \gamma_{k+1}^2\mathbb{E}\|v(x_k) + Z_{k+1}\|^2 + \gamma_{k+1}\mathbb{E}\|\sigma(x_k)\xi_{k+1}\|^2 + 2\gamma_{k+1}\mathbb{E}\langle x_k, v(x_k) + Z_{k+1}\rangle \\
&\quad + 2\gamma_{k+1}^{1/2}\mathbb{E}\langle x_k, \sigma(x_k)\xi_{k+1}\rangle + 2\gamma_{k+1}^{3/2}\mathbb{E}\langle v(x_k) + Z_{k+1}, \sigma(x_k)\xi_{k+1}\rangle \\
&\le 2L^2\gamma_{k+1}^2 a_k + 2\gamma_{k+1}^2\mathbb{E}\|Z_{k+1}\|^2 + \gamma_{k+1}C_\sigma + 2\gamma_{k+1}C_v(\sqrt{a_k}+1) + 2\gamma_{k+1}\sqrt{a_k}\sqrt{\mathbb{E}\|Z_{k+1}\|^2} \\
&\quad + 2\gamma_{k+1}^{3/2}\sqrt{C_\sigma}\sqrt{\mathbb{E}\|Z_{k+1}\|^2}
\end{aligned}
\tag{16}
$$

By Assumption 3, there is some $C_b > 0$ such that $\mathbb{E}\|b_{k+1}\|^2 \le C_b(\gamma_{k+1}^2 a_k + \gamma_{k+1})$, and we have

$$
\mathbb{E}\|Z_{k+1}\|^2 \le 2\mathbb{E}\|b_{k+1}\|^2 + 2\mathbb{E}\|U_{k+1}\|^2 \le 2C_b(\gamma_{k+1}^2 a_k + \gamma_{k+1}) + 2\sigma^2.
\tag{17}
$$

Moreover, as $\sqrt{p+q} \le \sqrt{p} + \sqrt{q}$, we have

$$
\sqrt{\mathbb{E}\|Z_{k+1}\|^2} \le \sqrt{2C_b}(\gamma_{k+1}\sqrt{a_k} + \sqrt{\gamma_{k+1}}) + \sqrt{2}\sigma.
\tag{18}
$$

Plugging the bounds from (17) and (18) into (16) gives

$$
\begin{aligned}
a_{k+1} - a_k &\le 2L^2\gamma_{k+1}^2 a_k + 4C_b\gamma_{k+1}^4 a_k + 4C_b\gamma_{k+1}^3 + 4\gamma_{k+1}^2\sigma^2 \\
&\quad + \gamma_{k+1}C_\sigma + 2\gamma_{k+1}C_v\sqrt{a_k} + 2\gamma_{k+1}C_v \\
&\quad + 2\sqrt{2C_b}\gamma_{k+1}^2 a_k + 2\sqrt{2C_b}\gamma_{k+1}^{3/2}\sqrt{a_k} + 2\sqrt{2}\sigma\gamma_{k+1}\sqrt{a_k} \\
&\quad + 2\sqrt{2C_bC_\sigma}\gamma_{k+1}^{5/2}\sqrt{a_k} + 2\sqrt{2C_bC_\sigma}\gamma_{k+1}^2 + 2\gamma_{k+1}^{3/2}\sqrt{2C_\sigma}\sigma \\
&=: P\gamma_{k+1}^2 a_k + Q\gamma_{k+1}\sqrt{a_k} + R\gamma_{k+1},
\end{aligned}
\tag{19}
$$

where

$$
\begin{aligned}
P &= 2L^2 + 4C_b\gamma_{k+1}^2 + 2\sqrt{2C_b} \\
Q &= 2C_v + 2\sqrt{2C_b}\sqrt{\gamma_{k+1}} + 2\sqrt{2}\sigma + 2\sqrt{2C_b}\gamma_{k+1} + 2\sqrt{2C_bC_\sigma}\gamma_{k+1}^{3/2} \\
R &= 4C_b\gamma_{k+1}^2 + 4\gamma_{k+1}\sigma^2 + C_\sigma + 2C_v + 2\sqrt{2C_bC_\sigma}\gamma_{k+1} + 2\gamma_{k+1}^{1/2}\sqrt{2C_\sigma}\sigma.
\end{aligned}
$$

The exact values of $P$, $Q$, and $R$ are irrelevant, and we only need upper bounds for them. Assuming that $\gamma_{k+1} < 1$ for all $k$, we replace the three quantities by

$$
\begin{aligned}
P &= 2L^2 + 4C_b + 2\sqrt{2C_b} \\
Q &= 2C_v + 2\sqrt{2C_b} + 2\sqrt{2}\sigma + 2\sqrt{2C_b} + 2\sqrt{2C_bC_\sigma} \\
R &= 4C_b + 4\sigma^2 + C_\sigma + 2C_v + 2\sqrt{2C_bC_\sigma} + 2\sqrt{2C_\sigma}\sigma.
\end{aligned}
\tag{20}
$$

Now, define $h_k = \gamma_{k+1}^2 a_k$. The recursion (19) in terms of $h_k$ becomes

$$
h_{k+1} \le h_k(1 + P\gamma_{k+1}^2)\frac{\gamma_{k+2}^2}{\gamma_{k+1}^2} + \sqrt{h_k}Q\gamma_{k+2}^2 + R\gamma_{k+1}\gamma_{k+2}^2.
$$

We now prove that there exists some $M > 0$ so that $h_k \le M\gamma_{k+1}$ by induction. Suppose it is the case for $k$, and we prove it for $k+1$. Using the induction hypothesis we get

$$
\begin{aligned}
h_{k+1} &\le M\gamma_{k+1}(1 + P\gamma_{k+1}^2)\frac{\gamma_{k+2}^2}{\gamma_{k+1}^2} + \sqrt{M\gamma_{k+1}}Q\gamma_{k+2}^2 + R\gamma_{k+1}\gamma_{k+2}^2 \\
&= M(1 + P\gamma_{k+1}^2)\frac{\gamma_{k+2}^2}{\gamma_{k+1}} + \sqrt{M}\,Q\sqrt{\gamma_{k+1}}\gamma_{k+2}^2 + R\gamma_{k+1}\gamma_{k+2}^2
\end{aligned}
$$

For the last to be less than $M\gamma_{k+2}$, we have to verify

$$M(1 + P\gamma_{k+1}^2)\frac{\gamma_{k+2}}{\gamma_{k+1}} + \sqrt{M}Q\sqrt{\gamma_{k+1}}\gamma_{k+2} + R\gamma_{k+1}\gamma_{k+2} \le M$$

or equivalently,

$$M\left(\frac{\gamma_{k+2}}{\gamma_{k+1}} + P\gamma_{k+1}\gamma_{k+2} - 1\right) + \sqrt{M}Q\sqrt{\gamma_{k+1}}\gamma_{k+2} + R\gamma_{k+1}\gamma_{k+2} \le 0.$$

This is a quadratic equation in $\sqrt{M}$, and for this inequality to hold, we prove that the leading coefficient is negative, and the largest root is bounded above by some constant not depending on $n$.

Negativity of the leading coefficient is equivalent to

$$\frac{\gamma_{k+2}}{\gamma_{k+1}} + P\gamma_{k+1}\gamma_{k+2} < 1,$$

which is implied by our assumption on the step size.

The larger root of the equation is

$$\frac{\left(-4\gamma_{k+1}^2\gamma_{k+2}^2 PR + \gamma_{k+1}\gamma_{k+2}(\gamma_{k+2}Q^2 + 4R) - 4R\gamma_{k+2}^2\right)^{1/2} + \sqrt{\gamma_{k+1}}\gamma_{k+2}Q}{2(1 - \gamma_{k+1}\gamma_{k+2}P - \gamma_{k+2}/\gamma_{k+1})}$$

$$< \frac{\sqrt{\gamma_{k+1}}\gamma_{k+2}Q + \sqrt{R\gamma_{k+1}\gamma_{k+2}}}{(1 - \gamma_{k+1}\gamma_{k+2}P - \gamma_{k+2}/\gamma_{k+1})}$$

$$\le \frac{\sqrt{\gamma_{k+1}}\gamma_{k+1}Q + \sqrt{R}\gamma_{k+1}}{(1 - \gamma_{k+1}\gamma_{k+2}P - \gamma_{k+2}/\gamma_{k+1})}.$$

By our assumption on the step size that

$$\frac{\gamma_{k+2}}{\gamma_{k+1}} + P\gamma_{k+1}\gamma_{k+2} < 1 - \gamma_{k+1},$$

we get that the larger root is smaller than

$$\frac{\sqrt{\gamma_{k+1}}\gamma_{k+1}Q + \sqrt{R}\gamma_{k+1}}{\gamma_{k+1}} = \sqrt{\gamma_{k+1}}Q + \sqrt{R} < Q + \sqrt{R}.$$

Letting $M := Q + \sqrt{R}$ gives the desired result.

The second argument of the lemma follows from Assumption 3 and the first result of the lemma. ∎

**Lemma 4.** *For a vector valued function $g \in L^2(\mathbb{R}; \mathbb{R}^d)$, one has*

$$\left\|\int_0^s g(u)\,du\right\|^2 \le \left(\int_0^s \|g(u)\|\,du\right)^2 \le s\int_0^s \|g(u)\|^2\,du.$$

# D  Proofs for Section 5

## D.1  Proof of Theorem 3

For brevity, let us write $\mathcal{F}_k$ instead of $\mathcal{F}_{\tau_k}$. Opening up $\|x_{k+1}\|^2 = \|x_k + \gamma_{k+1}\{v(x_k) + Z_{k+1}\} + \sqrt{\gamma_{k+1}}\sigma(x_k)\xi_{k+1}\|^2$ and ignoring every term that is zero-mean under $\mathbb{E}[\cdot\,|\,\mathcal{F}_k]$, we get

$$\mathbb{E}[\|x_{k+1}\|^2\,|\,\mathcal{F}_k] = \mathbb{E}\Big[\|x_k\|^2 + 2\gamma_{k+1}\langle x_k, v(x_k) + Z_{k+1}\rangle$$

$$+ \gamma_{k+1}^2\|v(x_k) + Z_{k+1}\|^2 + \gamma_{k+1}\|\sigma(x_k)\xi_{k+1}\|^2 + 2\gamma_{k+1}^{\frac{3}{2}}\langle\sigma(x_k)\xi_{k+1}, b_{k+1}\rangle\,\Big|\,\mathcal{F}_k\Big]$$

$$\le \|x_k\|^2 + 2\gamma_{k+1}(\langle x_k, v(x_k)\rangle + C_\sigma/2) + 2\gamma_{k+1}^2\|v(x_k)\|^2$$

$$+ \mathbb{E}\Big[2\gamma_{k+1}^2\|Z_{k+1}\|^2 + 2\gamma_{k+1}\langle x_k, Z_{k+1}\rangle + 2\gamma_{k+1}^{\frac{3}{2}}\langle\sigma(x_k)\xi_{k+1}, b_{k+1}\rangle\,\Big|\,\mathcal{F}_k\Big]$$

$$\le \|x_k\|^2 + 2\gamma_{k+1}\left(\langle x_k, v(x_k)\rangle + C_\sigma/2 + \gamma_{k+1}^{\frac{1}{2}}C_\sigma/4\right) + 2\gamma_{k+1}^2\|v(x_k)\|^2 \qquad (21)$$

$$+ \mathbb{E}\big[2\gamma_{k+1}^2\|Z_{k+1}\|^2\,|\,\mathcal{F}_k\big] + \gamma_{k+1}^{\frac{3}{2}}\mathbb{E}\big[\|b_{k+1}\|^2\,|\,\mathcal{F}_k\big] + 2\mathbb{E}[\gamma_{k+1}\langle x_k, b_{k+1}\rangle|\mathcal{F}_k].$$

Recalling (5) in Assumption 3, we have for some $C > 0$

$$\mathbb{E}\|Z_{k+1}\|^2 \le 2\sigma^2 + 2C\left(\gamma_{k+1}^2 \mathbb{E}\|v(x_k)\|^2 + \gamma_{k+1}\right) \tag{22}$$

Without loss of generality, assume $\gamma_k \le 1$ and $\mathbb{E}\|x_k\|^2 \ge 1$ (so that $\left(\mathbb{E}\|x_k\|^2\right)^2 \ge \mathbb{E}\|x_k\|^2$) for all $k$. Then, $\|v(x_k)\|^2 \le L^2\|x_k\|^2$, together with Assumption 4 and the Cauchy-Schwartz inequality on the last term of (21), implies

$$
\begin{aligned}
\mathbb{E}\|x_{k+1}\|^2 &\le \mathbb{E}\|x_k\|^2 - 2\alpha\gamma_{k+1}\mathbb{E}\|x_k\|^2 + 2\gamma_{k+1}\left(\beta + C_\sigma + \frac{1}{2}\gamma_{k+1}^{\frac{1}{2}}C_\sigma\right) + 2L^2\gamma_{k+1}^2\mathbb{E}\|x_k\|^2 \\
&\quad + 2\gamma_{k+1}^2\left[2\sigma^2 + 2C\left(L^2\gamma_{k+1}^2\mathbb{E}\|x_k\|^2 + \gamma_{k+1}\right)\right] \\
&\quad + \gamma_{k+1}^{\frac{3}{2}}C\left(L^2\gamma_{k+1}^2\mathbb{E}\|x_k\|^2 + \gamma_{k+1}\right) \\
&\quad + 2\gamma_{k+1}\sqrt{C}\sqrt{L^2\gamma_{k+1}^2\left(\mathbb{E}\|x_k\|^2\right)^2 + \gamma_{k+1}\mathbb{E}\|x_k\|^2} \\
&\le \mathbb{E}\|x_k\|^2(1 - C_1\gamma_{k+1} + C_2\gamma_{k+1}^{\frac{3}{2}}) + C_3\gamma_{k+1}
\end{aligned}
$$

for some constants $C_1, C_2, C_3$ depending on $L, C, \sigma, \alpha, \beta$, and $d$. Since $\gamma_k \to 0$, there exist $\tilde{\alpha}, \tilde{\beta} > 0$ and $k_0$ such that, for all $k \ge k_0$,

$$\mathbb{E}\|x_{k+1}\|^2 \le \mathbb{E}\|x_k\|^2(1 - \tilde{\alpha}\gamma_{k+1}) + \tilde{\beta}\gamma_{k+1}, \quad 1 - \tilde{\alpha}\gamma_{k+1} > 0.$$

A simple induction yields

$$\sup_k \mathbb{E}\|x_k\|^2 \le \max\left\{\frac{\tilde{\beta}}{\tilde{\alpha}}, \mathbb{E}\|x_{k_0}\|^2\right\}$$

which concludes the proof. $\blacksquare$

## D.2 Proof of Theorem 4 for Constant Diffusion

Before proceeding, we need a lemma which can be distilled from [20, Proposition 8]:

**Lemma 5.** *Suppose $\nabla f$ is $L$-Lipschitz. Fix $x \in \mathbb{R}^d$ and $\gamma > 0$, let $\tilde{x}^+ = x - \gamma\nabla f(x) + \sqrt{2\gamma}\xi$. Then*

$$\mathbb{E}\left[\exp\left(\frac{1}{2}\langle\nabla f(x), \tilde{x}^+ - x\rangle + \frac{L}{4}\|\tilde{x}^+ - x\|^2\right)\right] \le (1 - \gamma L)^{-d/2}e^{-\frac{\gamma}{4}\|\nabla f(x)\|^2}. \tag{23}$$

Let $\tilde{x}_{k+1} := x_k - \gamma_{k+1}\nabla f(x_k) + \sqrt{2\gamma_{k+1}}\xi_{k+1}$ so that $x_{k+1} - x_k = \tilde{x}_{k+1} - x_k - \gamma_{k+1}(U_{k+1} + b_{k+1})$. Conditioned on $x_k, U_{k+1}, U'_{k+1}, \xi'_{k+1}$, and using the $L$-Lipschitzness of $\nabla f$, we get

$$e^{-\frac{1}{2}f(x_k)}\mathbb{E}e^{\frac{1}{2}f(x_{k+1})}$$

$$\le \mathbb{E}\exp\left(\frac{1}{2}\langle\nabla f(x_k), x_{k+1} - x_k\rangle + \frac{L}{4}\|x_{k+1} - x_k\|^2\right) \tag{24}$$

$$\le \mathbb{E}\exp\left\{\frac{1}{2}\langle\nabla f(x_k), \tilde{x}_{k+1} - x_k\rangle - \frac{1}{2}\langle\nabla f(x_k), \gamma_{k+1}U_{k+1}\rangle \right. \tag{25}$$

$$\left. - \frac{1}{2}\langle\nabla f(x_k), \gamma_{k+1}b_{k+1}\rangle + \frac{L}{2}\|\tilde{x}_{k+1} - x_k\|^2 + L\gamma_{k+1}^2\|U_{k+1}\|^2 + L\gamma_{k+1}^2\|b_{k+1}\|^2\right\}. \tag{26}$$

Let $\delta \in (0, 1)$. Since

$$-\frac{1}{2}\langle\nabla f(x_k), \gamma_{k+1}U_{k+1}\rangle \le \gamma_{k+1}^{2-\delta}\|\nabla f(x_k)\|^2 + \gamma_{k+1}^\delta\|U_{k+1}\|^2,$$

$$-\frac{1}{2}\langle\nabla f(x_k), \gamma_{k+1}b_{k+1}\rangle \le \gamma_{k+1}^2\|\nabla f(x_k)\|^2 + \|b_{k+1}\|^2,$$

we have

$$e^{-\frac{1}{2}f(x_k)}\mathbb{E}e^{\frac{1}{2}f(x_{k+1})} \tag{27}$$

$$\le \mathbb{E}\exp\left\{\frac{1}{2}\langle\nabla f(x_k),\tilde{x}_{k+1}-x_k\rangle + \frac{L}{2}\|\tilde{x}_{k+1}-x_k\|^2\right. \tag{28}$$

$$\left.+\left(\gamma_{k+1}^{2-\delta}+\gamma_{k+1}^2\right)\|\nabla f(x_k)\|^2 + \left(L\gamma_{k+1}^2+\gamma_{k+1}^\delta\right)\|U_{k+1}\|^2 + \left(L\gamma_{k+1}^2+1\right)\|b_{k+1}\|^2\right\}. \tag{29}$$

Invoking (11) an denoting $c' \triangleq \left(L\gamma_{k+1}^2+1\right)\cdot c$, we get

$$e^{-\frac{1}{2}f(x_k)}\mathbb{E}e^{\frac{1}{2}f(x_{k+1})} \le e^{A_k}\cdot\mathbb{E}\exp\left\{\frac{1}{2}\langle\nabla f(x_k),\tilde{x}_{k+1}-x_k\rangle + \frac{L}{2}\|\tilde{x}_{k+1}-x_k\|^2 + c'\cdot\gamma_{k+1}\|\xi_{k+1}\|^2\right\}, \tag{30}$$

where,

$$A_k \triangleq \left(\gamma_{k+1}^{2-\delta}+\gamma_{k+1}^2+c'\gamma_{k+1}^2\right)\|\nabla f(x_k)\|^2$$
$$+\left(L\gamma_{k+1}^2+\gamma_{k+1}^\delta\right)\|U_{k+1}\|^2 \tag{31}$$
$$+c'\left(\gamma_{k+1}^2\|U'_{k+1}\|^2+\gamma_{k+1}\|\xi'_{k+1}\|^2\right).$$

Recalling that $\sqrt{2\gamma_{k+1}}\xi_{k+1} = \tilde{x}_{k+1}-x_k+\gamma_{k+1}\nabla f(x_k)$, we have $\gamma_{k+1}\|\xi_{k+1}\|^2 \le \|\tilde{x}_{k+1}-x_k\|^2 + \gamma_{k+1}^2\|\nabla f(x_k)\|^2$, and thus

$$e^{-\frac{1}{2}f(x_k)}\mathbb{E}e^{\frac{1}{2}f(x_{k+1})} \le e^{A'_k}\cdot\mathbb{E}\exp\left\{\frac{1}{2}\langle\nabla f(x_k),\tilde{x}_{k+1}-x_k\rangle + \left(\frac{L}{2}+c'\right)\|\tilde{x}_{k+1}-x_k\|^2\right\}, \tag{32}$$

where $A'_k = A_k + c'\gamma_{k+1}^2\|\nabla f(x_k)\|^2$. Lemma 5 then implies

$$e^{-\frac{1}{2}f(x_k)}\mathbb{E}e^{\frac{1}{2}f(x_{k+1})} \le e^{A''_k}\cdot(1-\gamma_{k+1}L')^{-\frac{d}{2}} \tag{33}$$

where $A''_k = A'_k - \frac{\gamma_{k+1}}{4}\|\nabla f(x_k)\|^2$.

We now take the expectation over $x_k, U_{k+1}, U'_{k+1}, \xi'_{k+1}$ (in other words, we are now only conditioning on $x_k$). Set $\epsilon \triangleq (1-\gamma_{k+1}L')^{-\frac{1}{2}}-1 > 0$. Since $U_{k+1}, U'_{k+1}, \xi'_{k+1}$ are sub-Gaussian and since $\gamma_k \to 0$, for $k$ sufficiently large we have

$$\mathbb{E}A''_k \le (1+\epsilon)\cdot\exp\left[\left(-\frac{\gamma_{k+1}}{4}+\gamma_{k+1}^{2-\delta}+\gamma_{k+1}^2+c'\gamma_{k+1}^2+c'\gamma_{k+1}^2\right)\|\nabla f(x_k)\|^2\right] \tag{34}$$

$$\le (1+\epsilon)\cdot e^{-\frac{\gamma_{k+1}}{8}\|\nabla f(x_k)\|^2}. \tag{35}$$

To summarize, we have shown that, conditioned on $x_k$,

$$e^{-\frac{1}{2}f(x_k)}\mathbb{E}e^{\frac{1}{2}f(x_{k+1})} \le (1-\gamma_{k+1}L')^{-\frac{d+1}{2}}e^{-\frac{\gamma_{k+1}}{8}\|\nabla f(x_k)\|^2}. \tag{36}$$

A simple induction à la [20, Lemma 1 & Proposition 8] then concludes the proof. ∎

## D.3 Proof of Theorem 4 for Mirror Langevin

Here, we bring the proof of Theorem 4 for the case of Example 4 and without noise. The proof for the noisy case is the same as in Appendix D.2.

Define

$$x^+ = x - \gamma\nabla f\circ\nabla\phi^*(x) + \sqrt{2\gamma}(\nabla^2\phi^*(x)^{-1})^{1/2}\xi,$$

where $\xi$ is a standard Gaussian random variable. Let $U(x) = f(\nabla\phi^*(x))$. For a fixed $x$, we have

$$\mathbb{E}e^{\frac{1}{2}U(x^+)-\frac{1}{2}U(x)} = \frac{1}{(2\pi)^{d/2}}\int\exp\left(\frac{1}{2}U(x^+)-\frac{1}{2}U(x)-\frac{\|\xi\|^2}{2}\right)d\xi$$

Notice that we have

$$\xi = \frac{1}{\sqrt{2\gamma}}(\nabla^2\phi^*(x))^{1/2}\big(x^+ - x + \gamma\nabla f \circ \nabla\phi^*(x)\big)$$

which implies

$$d\xi = (\sqrt{2\gamma})^{-d}\sqrt{\det\nabla^2\phi^*(x)}\,dx^+$$

Thus, the integral, after the change of variable from $\xi$ to $x^+$ becomes

$$\frac{1}{C}\int\exp\left(\frac{1}{2}U(x^+) - \frac{1}{2}U(x) - \frac{1}{4\gamma}\|(\nabla^2\phi^*(x))^{1/2}\big(x^+ - x + \gamma\nabla f \circ \nabla\phi^*(x)\big)\|^2\right)dx^+ \qquad (37)$$

with $C = (4\pi\gamma)^{d/2}\sqrt{\det\nabla^2\phi^*(x)^{-1}}$. Now we use the smoothness of $f$:

$$U(x^+) - U(x) = f(\nabla\phi^*(x^+)) - f(\nabla\phi^*(x))$$

$$\leq \langle\nabla^2\phi^*(x)\nabla f(\nabla\phi^*(x)), x^+ - x\rangle + \frac{L}{2}\|x^+ - x\|^2$$

On the other hand, we have

$$\|(\nabla^2\phi^*(x))^{1/2}\big(x^+ - x + \gamma\nabla f \circ \nabla\phi^*(x)\big)\|^2$$

$$= \|(\nabla^2\phi^*(x))^{1/2}(x^+ - x)\|^2 + \gamma^2\|(\nabla^2\phi^*(x))^{1/2}\nabla f(\nabla\phi^*(x))\|^2$$

$$+ 2\gamma\langle\nabla^2\phi^*(x)\nabla f\nabla\phi^*(x), x^+ - x\rangle$$

Notice that in (37), the colored terms cancel out, and what we are left with is

$$\mathbb{E}e^{\frac{1}{2}U(x^+) - \frac{1}{2}U(x)}$$

$$\leq \frac{1}{C}\int\exp\left(\frac{L}{4}\|x^+ - x\|^2 - \frac{1}{4\gamma}\|(\nabla^2\phi^*(x))^{1/2}(x^+ - x)\|^2 - \frac{\gamma}{4}\|(\nabla^2\phi^*(x))^{1/2}\nabla f(\nabla\phi^*(x))\|^2\right)dx^+$$

As, by our assumption, $\nabla^2\phi^*$ is bounded from above and below, we get the exact form as in Lemma 5. The rest of the proof is the same as in Appendix D.2. ∎

## D.4 Proof of Proposition 1

In this section, we prove that Examples 1–6 satisfy our bias conditions, which, as we have seen in Section 5, implies Proposition 1. For brevity, we write $\mathscr{F}_k$ for $\mathscr{F}_{\tau_k}$.

§ **Proof for Example 1.** For randomized mid-point method, by replacing $\widetilde{\nabla}f(x_k)$ and $\widetilde{\nabla}f(x_{k+1/2})$ with $\nabla f(x_k) + U'_{k+1}$ and $\nabla f(x_{k+1/2}) + U_{k+1}$ respectively, we have

$$x_{k+1/2} = x_k - \gamma_{k+1}\alpha_{k+1}\{\nabla f(x_k) + U'_{k+1}\} + \sqrt{2\gamma_{k+1}\alpha_{k+1}}\xi'_{k+1},$$

$$x_{k+1} = x_k - \gamma_{k+1}\{\nabla f(x_{k+1/2}) + U_{k+1}\} + \sqrt{2\gamma_{k+1}}\xi_{k+1},$$

where $\{\alpha_k\}$ are i.i.d. and uniformly distributed in $[0, 1]$, $\{U_k\}$ and $\{U'_k\}$ are noises in evaluating $\nabla f$ at the corresponding points, and $\xi_k, \xi'_k$ are independent standard Gaussians.

Notice that the Lipschitzness of $\nabla f$, and the fact that $\alpha_k \leq 1$ implies that the bias term $b_{k+1} := \nabla f(x_{k+1/2}) - \nabla f(x_k)$ satisfies

$$\mathbb{E}[\|b_{k+1}\|^2\,|\,\mathscr{F}_k] \leq L^2\mathbb{E}[\|x_{k+1/2} - x_k\|^2\,|\,\mathscr{F}_k]$$

$$\leq L^2\Big(\gamma_{k+1}^2\mathbb{E}[\|\nabla f(x_k) + U'_{k+1}\|^2\,|\,\mathscr{F}_k] + 2\gamma_{k+1}d\Big)$$

$$\leq 2L^2\gamma_{k+1}^2\|\nabla f(x_k)\|^2 + 2L^2\gamma_{k+1}^2\sigma^2 + 2L^2d\gamma_{k+1}$$

$$= O(\gamma_{k+1}^2\|\nabla f(x_k)\|^2 + \gamma_{k+1}).$$

**§ Proof for Example 2.**   Recall that the new algorithm Optimistic Randomized Mid-Point Method has the iterates

$$x_{k+1/2} = x_k - \gamma_{k+1}\alpha_{k+1}\widetilde{\nabla} f(x_{k-\frac{1}{2}}) + \sqrt{2\gamma_{k+1}\alpha_{k+1}}\,\xi'_{k+1},$$

$$x_{k+1} = x_k - \gamma_{k+1}\widetilde{\nabla} f(x_{k+1/2}) + \sqrt{2\gamma_{k+1}}\,\xi_{k+1},$$

where $\{\alpha_k\}$, $\xi_k, \xi'_k$, and $\widetilde{\nabla} f$ are the same as in (RMM), and the noise and bias are $U_{k+1} :=$ $\widetilde{\nabla} f(x_{k+1/2}) - \nabla f(x_{k+1/2})$ and $b_{k+1} := \nabla f(x_{k+1/2}) - \nabla f(x_k)$. We have

$$
\begin{aligned}
\mathbb{E}[\|b_{k+1}\|^2 \mid \mathscr{F}_k] &= \mathbb{E}[\|\nabla f(x_{k+1/2}) - \nabla f(x_k)\|^2 \mid \mathscr{F}_k] \\
&\le L^2 \mathbb{E}[\|x_{k+1/2} - x_k\|^2 \mid \mathscr{F}_k] \\
&= L^2 \mathbb{E}[\|-\gamma_{k+1}\alpha_{k+1}\widetilde{\nabla} f(x_{k-\frac{1}{2}}) + \sqrt{2\gamma_{k+1}\alpha_{k+1}}\,\xi'_{k+1}\|^2 \mid \mathscr{F}_k] \\
&\le 2L^2\gamma_{k+1}^2 \mathbb{E}[\|\nabla f(x_{k-\frac{1}{2}})\| \mid \mathscr{F}_k] + 2L^2\gamma_{k+1}^2\sigma^2 + 4L^2 d\gamma_{k+1}.
\end{aligned}
$$

Similar to the proof for Example 6, notice that $\|\nabla f(x_{k-\frac{1}{2}})\|^2 \le 2\|\nabla f(x_{k-\frac{1}{2}}) - \nabla f(x_k)\|^2 + 2\|\nabla f(x_k)\|^2$. As $\gamma_k \to 0$, one can assume that $2L^2\gamma_{k+1}^2 < \frac{1}{2}$, and we get

$$\mathbb{E}[\|b_{k+1}\|^2 \mid \mathscr{F}_k] \le 4L^2\gamma_{k+1}^2\|\nabla f(x_k)\|^2 + 4L^2\gamma_{k+1}^2\sigma^2 + 8L^2 d\gamma_{k+1} = O(\gamma_{k+1}^2\|\nabla f(x_k)\|^2 + \gamma_{k+1}),$$

as desired.  ∎

**§ Proof for Example 3.**   The iterates of stochastic Runge-Kutta Langevin algorithm is as follows:

$$h_1 = x_k + \sqrt{2\gamma_{k+1}}\Big[(1/2 + 1/\sqrt{6})\,\xi_{k+1} + \xi'_{k+1}/\sqrt{12}\Big]$$

$$h_2 = x_k - \gamma_{k+1}\{\nabla f(x_k) + U'_{k+1}\} + \sqrt{2\gamma_{k+1}}\Big[(1/2 - 1/\sqrt{6})\,\xi_{k+1} + \xi'_{k+1}/\sqrt{12}\Big]$$

$$x_{k+1} = x_k - \frac{\gamma_{k+1}}{2}(\nabla f(h_1) + \nabla f(h_2)) + \gamma_{k+1}U_{k+1} + \sqrt{2\gamma_{k+1}}\,\xi_{k+1},$$

where $\xi_{k+1}$ and $\xi'_{k+1}$ are independent standard Gaussian random variables independent of $x_k$, and $U_{k+1}$ and $U'_{k+1}$ are noise in the evaluation of $f$.

Observe that

$$b_{k+1} = \frac{1}{2}(\nabla f(h_1) - \nabla f(x_k)) + \frac{1}{2}(\nabla f(h_2) - \nabla f(x_k)).$$

We have

$$\mathbb{E}[\|\nabla f(h_1) - \nabla f(x_k)\|^2 \mid \mathscr{F}_k] \le 2L^2 d(1/4 + 1/6 + 1/12)\gamma_{k+1} = O(\gamma_{k+1}),$$

and

$$
\begin{aligned}
\mathbb{E}[\|\nabla f(h_2) - \nabla f(x_k)\|^2 \mid \mathscr{F}_k] &\le 2L^2\Big(\gamma_{k+1}^2\|\nabla f(x_k)\|^2 + 2\gamma_{k+1}^2\sigma^2 + 2d(1/4 - 1/6 + 1/12)\gamma_{k+1}\Big) \\
&= O(\gamma_{k+1}^2\|\nabla f(x_k)\|^2 + \gamma_{k+1}).
\end{aligned}
$$

We thus have

$$
\begin{aligned}
\mathbb{E}[\|b_{k+1}\|^2 \mid \mathscr{F}_t] &\le \frac{1}{2}\mathbb{E}[\|\nabla f(h_1) - \nabla f(x_k)\|^2 \mid \mathscr{F}_k] + \frac{1}{2}\mathbb{E}[\|\nabla f(h_2) - \nabla f(x_k)\|^2 \mid \mathscr{F}_k] \\
&= O(\gamma_{k+1}^2\|\nabla f(x_k)\|^2 + \gamma_{k+1}),
\end{aligned}
$$

as desired.  ∎

**§ Proof for Example 4.**   Suppose $\phi$ is a Legendre function [52] for $\mathbb{R}^d$, and consider the iterates

$$x_{k+1} = x_k - \gamma_{k+1}\nabla f(\nabla \phi^*(x_k)) + \sqrt{2\gamma_{k+1}}(\nabla^2\phi^*(x_k)^{-1})^{1/2}\,\xi_{k+1},$$

where $\phi^*$ is the *Fenchel dual* of $\phi$, that is, $\phi^*(x) = \sup_{y \in \mathbb{R}^d}(\langle x, y\rangle - \phi(y))$. Also recall that [52]

$$\nabla\phi(\nabla\phi^*(x)) = x, \quad \nabla^2\phi^*(\nabla\phi(x))^{-1} = \nabla^2\phi(x), \quad \forall x \in \mathbb{R}^d.$$

Let $v = -\nabla f \circ \nabla\phi^*$ and $\sigma = (\nabla^2\phi^*)^{-1/2}$. First, we mention what our assumptions imply on $f$:

- The Lipschitzness of $v$ corresponds to a similar condition in [31, A2]:

$$\|\nabla f(x) - \nabla f(y)\| \le L\|\nabla\phi(x) - \nabla\phi(y)\|$$

- The Lipschitzness of $\sigma$ in Frobenius norm corresponds to *modified self-concordance* in [31, A1]:

$$\|\nabla^2\phi(x)^{1/2} - \nabla^2\phi(y)^{1/2}\|_F \le L\|\nabla\phi(x) - \nabla\phi(y)\|.$$

- Boundedness of $\sigma$ in Hilbert-Schmidt norm implies

$$\left\|\nabla^2\phi(x)^{-1/2}\right\|_F \le C_\sigma.$$

- Dissipativity and weak-dissipativity of $v$ corresponds to the conditions below, respectively:

$$\langle\nabla\phi(x), \nabla f(x)\rangle \ge \alpha\|\nabla\phi(x)\|^2 - \beta, \quad \langle\nabla\phi(x), \nabla f(x)\rangle \ge \alpha\|\nabla\phi(x)\|^{1+\kappa} - \beta.$$

If $f$ and $\phi$ satisfy the conditions above, then the mirror Langevin algorithm Example 4 fits into the (LRM) scheme.

*Remark.* Note that this version of Mirror Langevin *cannot* handle the case where $e^{-f}$ is supported on a compact domain; in that case, the Hessian of $\phi$ *has to* blow up near the boundary, and will not satisfy our boundedness assumption. The version of mirror Langevin we consider in this paper, though, can be thought as an adaptive conditioning method for densities supported on $\mathbb{R}^d$. This setting has also been studied in the literature, see [55].

## § Proof for Example 6.   The iterates of (PLA) follow

$$x_{k+1} = x_k - \gamma_{k+1}\nabla f(x_{k+1}) + \sqrt{2\gamma_{k+1}}\,\xi_{k+1}. \tag{PLA}$$

We mentioned that the bias term is $b_{k+1} = \nabla f(x_{k+1}) - \nabla f(x_k)$. Now it remains to prove that it satisfies the conditions (5) and (11). We have

$$
\begin{aligned}
\mathbb{E}[\|b_{k+1}\|^2 \mid \mathscr{F}_k] &= \mathbb{E}[\|\nabla f(x_{k+1}) - \nabla f(x_k)\|^2 \mid \mathscr{F}_k] \\
&\le L^2\mathbb{E}[\|x_{k+1} - x_k\|^2 \mid \mathscr{F}_k] \\
&= L^2\mathbb{E}[\|-\gamma_{k+1}\nabla f(x_{k+1}) + \sqrt{2\gamma_{k+1}}\,\xi_{k+1}\|^2 \mid \mathscr{F}_k] \\
&\le 2L^2\gamma_{k+1}^2\mathbb{E}[\|\nabla f(x_{k+1})\|^2 \mid \mathscr{F}_k] + 4L^2 d\gamma_{k+1}.
\end{aligned}
$$

Now, notice that $\|\nabla f(x_{k+1})\|^2 \le 2\|\nabla f(x_{k+1}) - \nabla f(x_k)\|^2 + 2\|\nabla f(x_k)\|^2$. As $\gamma_k \to 0$, one can assume that $2L^2\gamma_{k+1}^2 < \frac{1}{2}$, and we get

$$\mathbb{E}[\|b_{k+1}\|^2 \mid \mathscr{F}_k] \le \frac{1}{2}\mathbb{E}[\|b_{k+1}\|^2 \mid \mathscr{F}_k] + \|\nabla f(x_k)\|^2 + 4L^2 d\gamma_{k+1},$$

which implies

$$\mathbb{E}[\|b_{k+1}\|^2 \mid \mathscr{F}_k] \le 4L^2\gamma_{k+1}^2\|\nabla f(x_k)\|^2 + 8L^2 d\gamma_{k+1} = O(\gamma_{k+1}^2\|\nabla f(x_k)\|^2 + \gamma_{k+1}),$$

as desired. ∎

