# OpenReview forum: "A Dynamical System View of Langevin-Based Non-Convex Sampling"
_NeurIPS.cc/2023/Conference — NeurIPS 2023 spotlight_

### Official Review · Reviewer_3YTY · 2023-07-01

**Soundness:** 3 good
**Presentation:** 3 good
**Contribution:** 3 good
**Rating:** 7
**Confidence:** 3

**Summary:**

This paper presents a general framework for studying the convergence of last-iterate, noisy, and possibly biased Langevin-like discrete time approximations to the continuous Langevin flow for sampling from a distribution.

**Strengths:**

This paper is (with one small quibble which I will explain below) very well written and well explained. Using the machinery of asymptotic pseudotrajectories, it establishes asymptotic convergence of a wide range of discrete time sampling schemes to the Boltzman distribution $e^{-f}$. The advantage of the pseudotrajectory framework is that it replaces standard "Cauchy-type" convergence (which requires checking that an infinitely long tail of iterates converge) with a weaker notion of convergence that only requires "finite length" tails.

The upshot is that, due to cited source [6], this weaker sense of convergence is sufficient for convergence to a limit point of the original flow $\Phi$, provided that $\Phi$ has a unique fixed point. This allows for much more freedom in the analysis, and hence encompasses a wider range of conditions. The technique also has advantages of discretization-based approaches as explained by the author.

**Weaknesses:**

My biggest complain with the paper is that perhaps the most interesting part of the framework is the 2-line proof of Theorem 2 relegated to the appendix. This argument relies on  the limit-set characterization of [6] to show that asymptotic pseudotrajectories are sufficient to ensure convergence to the fixed point of the Langevin flow, namely the correct Boltzman distribution. I think the authors should explain this point in the body, and explain which theorem of [6] is being invoked (I suspect Thm 0.1), and how it's conditions are met. That way, the reader understands not simply how the proof framework of the paper works, but why we should believe that is somehow the "correct" one.

**Questions:**

What is the theorem of [6] being invoked here? Can the authors explain its conditions and why they are met.

**Limitations:**

This work is purely asymptotic, and provides no rates of convergence.

---

> ### Author Rebuttal · Authors · 2023-08-07
>
> Thank you for your input and remarks. We reply to your questions below, and we will revise our manuscript accordingly in the upcoming revision.
>
>
>
> > My biggest complain with the paper is that perhaps the most interesting part of the framework is the 2-line proof of Theorem 2 relegated to the appendix. This argument relies on the limit-set characterization of [6] to show that asymptotic pseudotrajectories are sufficient to ensure convergence to the fixed point of the Langevin flow, namely the correct Boltzman distribution. I think the authors should explain this point in the body, and explain which theorem of [6] is being invoked (I suspect Thm 0.1), and how it's conditions are met. That way, the reader understands not simply how the proof framework of the paper works, but why we should believe that is somehow the "correct" one.
>
> We agree with the reviewer. We only made this choice due to the page limit. We will expand the proof of the theorem to include more details. Indeed, Theorem 0.1 (see Theorem 5.7 (i) in M. Benaïm (2006), "Dynamics of stochastic approximation algorithms" for an extended version) is used.
>
> > What is the theorem of [6] being invoked here? Can the authors explain its conditions and why they are met.
>
> Theorem 0.1 is used here. The assumptions are:
>
> - $(\mathrm{law}(X_t))_t$ is an APT in the corresponding metric space (which we prove in Theorem 1 in the paper)
> - $(\mathrm{law}(X_t))_t$ is precompact (which we prove in Theorem 2 that it is equivalent to Assumption (10))
> - $\Phi$ is integrable (which is implied by Assumption 1 in our paper)
>
> These imply that the limit-set of $(\mathrm{law}(X_t))_t$ is an *internally chain transitive* set.
>
> These conditions are met becuase Assumption (10) implies that the trajectory $(X_t)$ is precompact in the Wasserstein space. If the trajectory is also a Wasserstein APT, then it is guaranteed to converge to an ICT set of the flow corresponding to the SDE. For the case of the Langevin SDE, we can show that the ICT set is the singleton $\{ \pi \}$. Following the notation of M. Benaïm (2006), "Dynamics of stochastic approximation algorithms" section 6.2, let $M$ to be the set of absolutely continuous probability measures in $W_2$, $\Lambda = \{\pi\}$ and define $V(\mu) = D_{\mathrm{KL}}(\mu \mid \pi)$. Then, it is clear that $V$ is a Lyapunov function of the Langevin dynamics, whose value is strictly decreasing along the flow (as the time derivative of V along the flow is negative of the relative Fisher information, which is strictly positive for all measures other than $\pi$). Thus, all requirements of Proposition 6.4 are satisfied, showing that every ICT set is contained in $\Lambda$. In another words, the only point in the ICT set is $\pi$.
>
> ---
>
> We hope that the above addresses your questions - but please let us know if any of the above is not sufficiently clear.
>
> Thank you again for your input and positive evaluation,
>
> The authors

---

> > ### Comment · Reviewer_3YTY · 2023-08-14
> > **Thank you!**
> >
> > I will likely maintain my score, but please add the two-line proof mentioned in my review to the body. And please include your exposition above in Theorem 5.7 (i) in M. Benaïm (2006) to the appendix.

---

> > > ### Author Response · Authors · 2023-08-15
> > >
> > > Thanks again for your time and your fruitful comments. We will definitely include what was discussed in the final revision.

---

### Official Review · Reviewer_Vq75 · 2023-07-06

**Soundness:** 3 good
**Presentation:** 4 excellent
**Contribution:** 3 good
**Rating:** 8
**Confidence:** 3

**Summary:**

The authors offer theoretical guarantees on the convergence of the last iterate of a very generic class of sampling methods to the stationary  distribution for a very large classe of sampling schemes in non-convex settings. This is achieved by showing that a large class of discrete sampling schemes can be mapped to continuous time ones and then to a Wasserstein asymptotic pseudo-trajectory. The distribution at large times of this process then converges to the stationary one under quite general assumptions on the Langevin noise and an opportune annealing scheme for the learning rate.

**Strengths:**

The paper addresses the interesting and actively researched problem of sampling non-convex, high dimensional densities. It explores a general framework that can be specialised to a wide range of sampling schemes, thereby providing a valuable guarantee for practical sampling scenarios.

Notably, the authors employ a novel proof technique, which, to the best of my knowledge, is unique in the context of sampling problems. This innovative approach contributes to the originality of the paper and distinguishes it from existing research in the field, which mostly study these problems as gradient flows.

Furthermore, the paper exhibits exceptional writing quality, being highly readable, well-structured, and accessible to a broad audience. These qualities enhance its overall impact and make it easier for readers to grasp the main idea.

Considering these strengths, I strongly believe that this paper is of high caliber. It offers significant contributions to the field, with its practical relevance, original proof technique, and excellent presentation.

**Weaknesses:**

I think this is a solid paper without major weaknesses, so I just have a minor remark.
The main result of this paper is showing that at very large times the sampling schemes converge to the desired distribution. It would be interesting to relax this condition and obtain bounds on the Wasserstein distance for a large (but finite) number of steps.

**Questions:**

1. Within this framework, is it possible to extract how the typical time to convergence scales with the size of the system? (for example in the sense of obtaining a lower bound on the number of steps T(epsilon) to obtain a distance between the distribution after T steps and the stationary one less than epsilon).
2. Related to the previous point: can you say something quantitative on the performance of ORMM beyond requiring less gradient calls?
3. Can you comment on the case where sigma(x_k)=1/beta_k in the definition of LRM, where beta_k is positively diverging as k increase?

**Limitations:**

This paper contains no applications or experiments, as it's expected by a paper of this kind.

---

> ### Author Rebuttal · Authors · 2023-08-07
>
> Thank you for your input and remarks. We reply to your questions below, and we will revise our manuscript accordingly in the upcoming revision.
>
>
>
> > The main result of this paper is showing that at very large times the sampling schemes converge to the desired distribution. It would be interesting to relax this condition and obtain bounds on the Wasserstein distance for a large (but finite) number of steps.
>
> From our analysis, a doubly exponential bound can be deduced: $\mathcal{W}_2$ error = $O\left(\frac{1}{\log\log n}\right)$. However, we believe that this loose bound offers no significant advantage over asymptotic convergence. Consequently, we have chosen to omit it. It is important to note that obtaining polynomial bounds is infeasible due to NP-hardness.
>
> We are also able to obtain results that are close in spirit to non-asymptotic results: Assuming step-sizes $\gamma_n = \Theta(n^{-1})$ and exponential convergence of continuous-time dynamics (which holds under, e.g., LSI), we can show that an LRM scheme converges in $\mathcal{W}^2_2$ at a rate of $O(n^{-1})$ under a warm-start condition, **under the same noise and bias conditions**. While we cannot estimate the duration of the burn-in phase, our argument holds for a wider range of stochastic and biased algorithms than in the current literature.
>
> > Within this framework, is it possible to extract how the typical time to convergence scales with the size of the system? (for example in the sense of obtaining a lower bound on the number of steps T(epsilon) to obtain a distance between the distribution after T steps and the stationary one less than epsilon).
>
> Lower bounds on the time complexity for sampling problems is a very hard problem. There has been some recent works and progress is being made for specific algorithms and simpler problem classes, see, e.g., Chatterji et al. (2021) "Oracle Lower Bounds for Stochastic Gradient Sampling Algorithms" and Chewi et al. (2023) "Query lower bounds for log-concave sampling".
>
> In our setup, the class of target distributions is rich enough to encompass NP-Hard problems, implying that the most difficult problems within this class require at least exponential time to solve.
>
> > Related to the previous point: can you say something quantitative on the performance of ORMM beyond requiring less gradient calls?
>
> Unfortunately, we are unable to provide such comparisons in theory, although we point out that the lack of such quantitative bounds for ORMM is not specific to our framework: It is present already in the context of purely **deterministic convex-concave optimization**, where optimistic methods originate, since their convergence rates do not surpass those of their non-optimistic counterparts. In most cases, the advantages of these methods are largely empirical.
>
> In this regard, our primary contribution lies in offering flexibility: By recycling past gradients, one can preserve the asymptotic convergence of randomized mid-point, demonstrating the potential to maintain favorable convergence behavior while saving 50% of the per-iteration costs.
>
>
> > Can you comment on the case where sigma(x_k)=1/beta_k in the definition of LRM, where beta_k is positively diverging as k increase?
>
> This is a very nice question. Diffusions with vanishing diffusion coefficient usually arise from simulated annealing processes. The problem with these time-inhomogeneous systems is that the flow (the $\Phi$ in the paper) is not going to be well-defined. We can make two possible comments here:
>
> 1. One can take $\Phi$ to be the flow of the resulting *ODE* when $\beta_k \to \infty$ (rather than the SDE), and the LRM scheme with the solutions of the ODE. This is a similar approach as in our reference [6] for "asymptotically autonomous with limit equation" (see section 2 of [6]). Our analysis goes through without any changes, and the result will be "$(X_t)$ will almost surely converge to an ICT set of the ODE; in the case of annealed Langevin dynamics as the main process, this means that $(X_t)$ converges to a *stationary point* of the potential." This result is, however, unsatisfactory, as one usually performs simulated annealing to converge to a **global optimum**.
>
> 2. One can convert the time-inhomogeneous system to a time-homogeneous system (by, e.g., introducing a new variable for tracking time). This dynamics will have a well-defined flow. However, this flow does not have any equilibria (as the time variable goes to infinity).
>
> We will definitely explore this direction in our future works.
>
>
> ---
> We hope that the above addresses your questions - but please let us know if any of the above is not sufficiently clear.
>
> Thank you again for your input and positive evaluation,
>
> The authors

---

> > ### Comment · Reviewer_Vq75 · 2023-08-13
> >
> > Thanks for the rebuttal.
> > The responses were detailed and insightful, and I am happy to keep the current score

---

> > > ### Author Response · Authors · 2023-08-16
> > >
> > > Thanks a lot for your time and encouraging comments.

---

### Official Review · Reviewer_vdDX · 2023-07-06

**Soundness:** 3 good
**Presentation:** 3 good
**Contribution:** 3 good
**Rating:** 5
**Confidence:** 3

**Summary:**

The work studies when a discretized Langevin dynamics under the Robbins-Monro-type stepsizes can converge to the Gibbs distribution. The paper obtains asymptotic results with very mild assumptions, and the framework not only includes Euler discretization, but many other sampling schemes as well, such as mirror Langevin, proximal, randomized mid-point and Runge-Kutta methods. The analysis builds upon constructing a continuous-time trajectory via interpolating the iterates, Wasserstein asymptotic pseudotrajectory and checking the stability condition by invoking the dynamical system theory.

**Strengths:**

The paper is built upon solid mathematical analysis and it is a nice contribution to the vast literature of Langevin algorithms in machine learning. The builds provides a unified framework for the asymptotic guarantees under the Robbins-Monro scheme.

**Weaknesses:**

Even though the mathematical theory is nice, the results may not have too much practical importance. The author(s) emphasized that this work is about asymptotic analysis instead of the non-asymptotic analysis. However, because the results are of asymptotic nature, it not clear to me what insights the results can provide concerning the schemes such as mirror Langevin, proximal, randomized mid-point and Runge-Kutta methods, because you only have asymptotic guarantees, it is impossible to use these results to compare these algorithms with more classic and basic Euler discretization of Langevin algorithms.

Also, I find the discussion of existing literature less satisfactory. Some of the claims and statements may not be that accurate. For example, on page 1, ``Existing guarantees suffer from the drawback of lacking guarantees for the last-iterates’’  ``the convergence is typically given on the averaged iterates instead of the more natural last iterates’’. To the best of my knowledge, there are numerous works on Langevin algorithms in the past decade and most of them are about last iterates guarantees in Wasserstein, KL or other distances; see e.g. Dalalyan and Karagulyan (2019), Dalalyan and Riou-Durand (2020), Ma et al. (2019), Ma et al. (2021), Raginsky et al. (2017), Gao et al. (2022). ``and little is known beyond the elementary schemes of stochastic gradient Langevin dynamics.’’ This is not accurate either. There have been many studies in the literature about underdamped Langevin, high-order Langevin, non-reversible Langevin and other variants of SGLD such as Dalalyan and Riou-Durand (2020), Hu et al. (2020), Ma et al. (2021), Mou et al. (2021), Gao et al. (2022).

**Questions:**

(1)	Page 4. ``The usual Lyapunov-type analysis for sampling algorithms focuses on bounding the change in relative entropy across iterations…” ``this makes the Lyapunov analysis applicable only to the simple Euler-Maruyama discreteization of (LD)’’ I am not too sure whether these two statements are accurate. Lyapunov functions are often used in analyis of Langevin algorithms, to show uniform bounds on the moments, e.g. Raginsky et al. (2017), in the coupling methods, e.g. Dalalyan and Riou-Durand (2020). It is definitely applicable beyond the Euler-Maruyama scheme, e.g. Dalalyan and Riou-Durand (2020) uses the disretization proposed in Cheng et al. (2018) to analyze kinetic Langevin dynamics.

(2)	One technical point I would like to see more discussions is that in equation (1) in your Definition 1, it is for fixed $T>0$. Actually the dependence on $T$ can be exponential in $T$, which is quite common for weak approximation error in the literature. However, in order for the Langevin algorithm to converge to the Gibbs distribution, one often needs uniform-in-time guarantees, and would you need $T\rightarrow\infty$ in order to obtain Theorem 2?

(3)	Theorem 2 is a very nice and clean result. But I am surprised that you only need assumption (10) which is an assumption on the discretized dynamics only. The reason I am asking is that it seems to me that Assumptions 1-3 alone do not guarantee that the continuous-time Langevin SDE has a unique stationary distribution. If Theorem 2 holds, that means assumption (10) can imply that the continuous-time Langevin SDE has a unique stationary distribution? The existence of $\pi$ is necessary for Theorem 2 to hold.

**Limitations:**

I did not see such discussions about limitations.

---

> ### Author Rebuttal · Authors · 2023-08-07
>
> We sincerely appreciate Reviewer vdDX's thoughtful criticisms. After a thorough reading, we believe that these critiques primarily stem from presentational issues, which we fully acknowledge exist and will commit to improving them following your suggestions.
>
> In light of this, we sincerely ask for a re-evaluation of our work based on the points addressed in our rebuttal. We are open to further discussions and eager to address any additional questions or concerns they may have.
>
> > Because the results are of asymptotic nature, it not clear to me what insights the results can provide such as mirror Langevin, proximal, randomized mid-point and Runge-Kutta methods.
>
> In our revised version, we intend to highlight the following two insights that our results offer to practitioners:
>
> 1. **Validating existing methods:** We first note that methods like mirror Langevin and randomized mid-point currently lack even asymptotic guarantees in fully non-convex scenarios, such as sampling from neural network-defined distributions. Our work fills this gap by offering the first solid justification for these schemes, supporting practitioners in utilizing these methods confidently.
>
> 2. **Facilitating new algorithm design:** Our work motivates novel sampling methods through a straightforward verification of **Assumptions 3**. An illustrative instance involves the randomized mid-point method and Runge-Kutta integrators, wherein a substantial 50% reduction in computation per iteration can be achieved without compromising convergence by simply recycling past gradients; see **Example 2**.
>
>     We do acknowledge the inherent limitations of our approach: The balance between the benefits of saving gradient oracles and potential drawbacks remains an open question, necessitating case-by-case practical evaluation. Nevertheless, our theory provides a flexible algorithmic design template that extends beyond the current literature's scope.
>
> > There are numerous works on Langevin algorithms in the past decade and most of them are about last iterates guarantees...
>
> We apologize for the confusion and would like to clarify our intended meaning concerning the absence of guarantees for **fully non-log-concave setups**. This refers to scenarios lacking assumptions of convexity or functional inequalities like LSIs or Poincaré. The references cited by the Reviewer fall within the latter classification: Dalalyan and Karagulyan (2019), Dalalyan and Riou-Durand (2020), and Ma et al. (2021) are centered around the (strongly-)log-concave context. Meanwhile, assumptions 1-3 in Raginsky et al. (2017) and assumption 1 in Gao et al. (2022) imply Poincaré inequalities.
>
> We express our gratitude to the Reviewer for bringing this point of confusion to our attention; we will make the necessary modifications accordingly.
>
> > On ``little is known beyond the elementary schemes''
>
> We would like to clarify that our reference to "elementary schemes" pertains specifically to discretizing the **Langevin diffusion** mentioned in equation (LD). While Reviewer vdDX has rightfully pointed out the existence of elementary discretization schemes for **other SDEs**, such as the underdamped or higher-order Langevin, our primary focus in this paper remains centered on the Langevin diffusion. [It is worth noting that our framework can be adapted to analyze these schemes.]
>
> As for the existing SGLD variants, which encompass variance reduction methods, it is important to highlight that their core emphasis continues to revolve around unbiased gradients.
>
> ### Q1
> We thank the Reviewer for raising this question, which again stems from our presentational issue: Our primary focus is the impact of **bias** in these analyses. As the Reviewer rightfully pointed out, handling noisy gradients' impact on entropy is possible, but bias effects are more complex for current Lyapunov analyses. Consequently, these analyses are unsuitable for application to schemes like the mirror Langevin approach. It is within this context that our framework offers an alternative proof methodology, bypassing the necessity to track any Lyapunov function.
>
>
> ### Q2
> This is a very nice question. Indeed, the dependence on $T$ is exponential (see the last equation in line 547). However, the defining property of being a WAPT is that for **any** fixed $T > 0$, as the beginning of the time window $[t, t+T]$ goes to infinity, the sup distance goes to zero; see **Definition 1**. Our analysis then shows that it suffices to have a **finite** (but arbitrary) $T$ in equation (1), and no uniform control in $T$ is required.
>
>
> ### Q3
> The uniqueness of the stationary distribution of the continuous-time Langevin diffusion can be established in various ways under our assumptions. One possible route, in line with our dynamical system analysis, is as follows.
>
> 1. Assumption 1 ensures that the continuous-time Langevin diffusion has strong solutions; this is standard. Denote the distribution at time $t$ by $\mu_t$.
>
> 2. Following the notation of M. Benaïm (1999), "Dynamics of stochastic approximation algorithms" section 6.2, let $M$ be the set of absolutely continuous probability measures in $W_2$, $\Lambda =$ {$\pi$} and define $V(\mu) = D_{\mathrm{KL}}(\mu \| \pi)$. Then, it is clear that $V$ is a Lyapunov function of the Langevin dynamics, whose value is strictly decreasing along the flow (as the time derivative of $V$ along the flow is negative of the relative Fisher information, which is strictly positive for all measures other than $\pi$).
>
> 3. Thus, all requirements of Proposition 6.4 are satisfied, showing that every limit set of $\{\mu_t\}_{t\geq 0}$ is contained in $\Lambda$. In other words, the only point possible limit point is $\pi$.
>
> ---
> We hope that the above addresses your questions - but please let us know if any of the above is not sufficiently clear.
>
> Thank you again for your input and thoughtful criticisms,
>
> The authors

---

> > ### Comment · Reviewer_vdDX · 2023-08-19
> >
> > Thanks for the very detailed response. I think some of your explanations really helped me to understand and appreciate your work. Even though the practical relevance of your work is still not that convincing to me, I do appreciate your work provide a unified framework for sampling a very general class of targets, which is a nice addition to the literature of Langevin algorithms. I will raise my score.

---

> > > ### Author Response · Authors · 2023-08-20
> > > **Official Comment by Authors**
> > >
> > > Thank you for the constructive criticism and the re-assessment; we promise to revise our manuscript to incorporate the discussion with the Reviewer.

---

### Official Review · Reviewer_hN2a · 2023-07-06

**Soundness:** 4 excellent
**Presentation:** 3 good
**Contribution:** 3 good
**Rating:** 7
**Confidence:** 4

**Summary:**

This paper gives a unified asymptotic analysis of a broad class of stochastic algorithms that encompasses several variants of the Langevin algorithm. In particular, it can handle issues of inexact gradients, bias, noise, and problems beyond gradient-based algorithms. The key technique is the introduction of an intermediate process, termed the *Picard process*, which fits between the iterates and the continuous-time

**Strengths:**

The main strength of this paper is the unified nature of the convergence results. The general framework encompasses a fairly large variety of Langevin-type algorithms, and likely a decent class of problems beyond Langevin algorithms. It also gives a clean, unified analysis.

**Weaknesses:**

My only major criticism is that the paper seems to overstate the novelty of the analytic methods. In particular, I have not seen the specific Picard process defined here, but several works introduce related processes that fit between the iterates and the continuous time process. This enables similar triangle-inequality based convergence proofs. For example (but not limited to):

Chau, Ngoc Huy, et al. "On stochastic gradient langevin dynamics with dependent data streams: The fully nonconvex case." SIAM Journal on Mathematics of Data Science 3.3 (2021): 959-986.

Bubeck, Sébastien, Ronen Eldan, and Joseph Lehec. "Sampling from a log-concave distribution with projected Langevin Monte Carlo." Discrete & Computational Geometry 59 (2018): 757-783.

Additionally, last iterate convergence guarantees are not particularly rare. Both works cited above give last-iterate bounds from corresponding stationary distributions. Many of the works citing these papers do as well.

On a minor note, there are some confusing notations.
* $b_k$ is used for the bias, but then gets re-defined in the proof of Lemma 2.
* Using $\sigma$ for both the diffusion matrix and the variance bound on the gradient noise is mildly confusing.
* At the end of the proof of Lemma 1, it should be the limit as $n\to\infty$, instead of $k\to \infty$

**Questions:**

* Can you give examples beyond Langevin-type algorithms for which you can apply the  method?
* Can you get some more quantitative guarantees beyond asymptotic convergence?

**Limitations:**

These are adequately addressed.

---

> ### Author Rebuttal · Authors · 2023-08-07
>
> We are sincerely grateful for pointing out the missing references and remarks. We reply to your questions below, and we will revise our manuscript accordingly in the upcoming revision.
>
> > My only major criticism is that the paper seems to overstate the novelty of the analytic methods. In particular, I have not seen the specific Picard process defined here, but several works introduce related processes that fit between the iterates and the continuous time process. This enables similar triangle-inequality based convergence proofs. For example (but not limited to): Chau, Ngoc Huy, et al. "On stochastic gradient langevin dynamics with dependent data streams: The fully nonconvex case." SIAM Journal on Mathematics of Data Science 3.3 (2021): 959-986. Bubeck, Sébastien, Ronen Eldan, and Joseph Lehec. "Sampling from a log-concave distribution with projected Langevin Monte Carlo." Discrete & Computational Geometry 59 (2018): 757-783.
>
> We agree with Reviewer hN2a's assessment and sincerely appreciate the references they have provided. We will duly update the paper and highlight our contributions, as well as acknowledge that similar ideas have been explored in prior works.
>
> What distinguishes our work from the existing literature is the advantage of generalizing the Picard process to encompass a vastly wider class of algorithms, specifically the Langevin-Robbins-Monro schemes. Moreover, the integration of the Picard process with the theory of asymptotic pseudo-trajectories plays a pivotal role in our analysis, and both of these aspects present original contributions.
>
> > Additionally, last iterate convergence guarantees are not particularly rare. Both works cited above give last-iterate bounds from corresponding stationary distributions. Many of the works citing these papers do as well.
>
> We agree that there are numerous results on last iterates for different settings and algorithms. What we intended to express is that little is known in the **generic non-log-concave setup** (without convexity or functional inequalities) and scenarios involving **biased discretization**. For example, Bubeck et al. (2018) is for log-concave target distributions, and  Chau et al. (2021) is for unbiased gradient estimates (see Eqn. (7) in their paper).
>
> To alleviate any misunderstandings, we are committed to revising our exposition on related work.
>
> > On a minor note, there are some confusing notations. b_k is used for the bias, but then gets re-defined in the proof of Lemma 2. Using sigma for both the diffusion matrix and the variance bound on the gradient noise is mildly confusing. At the end of the proof of Lemma 1, it should be the limit as n -> infty , instead of k -> infty.
>
> Thanks a lot for pointing this out. The end of the proof of Lemma 1 should be $n \to \infty$ as you mentioned. We will make the notation more succinct in the final version.
>
> > Can you give examples beyond Langevin-type algorithms for which you can apply the method?
>
> Essentially, our framework can be applied across a spectrum of continuous-time dynamics, such as Underdamped/Higher-order Langevin, Hamiltonian Monte Carlo, Neural SDEs, and more. In addition, the scope extends to diverse discretization methods for these dynamics. Examples of such include Euler-Maruyama, leap-frog, and symplectic integrators.
>
>
>
> > Can you get some more quantitative guarantees beyond asymptotic convergence?
>
> At the expense of stronger assumptions, yes: We are able to obtain results that are close in spirit to non-asymptotic results. Assuming step-sizes $\gamma_n = \Theta(n^{-1})$ and exponential convergence of continuous-time dynamics (which holds under, e.g., LSI), we can show that an LRM scheme converges in $\mathcal{W}^2_2$ at a rate of $O(n^{-1})$ under a warm-start condition, **under the same noise and bias conditions**. While we cannot estimate the duration of the burn-in phase, our argument holds for a wider range of stochastic and biased algorithms than in the current literature. However, as our primary focus lies in the generic setting where such assumptions are unavailable, we have chosen to defer these studies to future work.
>
>
> ---
>
> We hope that the above addresses your questions - but please let us know if any of the above is not sufficiently clear.
>
> Thank you again for your input and positive evaluation,
>
> The authors

---

> > ### Comment · Reviewer_hN2a · 2023-08-18
> > **Response**
> >
> > Thank you for your updates. As I mentioned, I am quite positive on this work. The main thing for me, is to place it into the context of existing work better, which I believe  you have  done.

---

> > > ### Author Response · Authors · 2023-08-20
> > > **Official Comment by Authors**
> > >
> > > Thank you for the expert's review. We also express our gratitude again for pointing out the missing link to prior work.

---

### Author Rebuttal · Authors · 2023-08-09

Dear AC and dear reviewers,

We wish to express our sincere gratitude for your dedicated efforts. Your insightful critiques and favorable evaluation have been acknowledged, and we have responded to all your inquiries in a detailed point-by-point manner, presented below.

After thorough consideration of your remarks, we wish to affirm the validity of the concerns highlighted by the reviewers. We firmly believe that these concerns can be aptly addressed through minor revisions, as detailed in our individual response. We appreciate your invaluable contributions towards refining our manuscript.


With utmost appreciation,

The authors

---

### Decision · Program_Chairs · 2023-09-21

**Decision:**

Accept (spotlight)

**Comment:**

This paper considers the convergence of discretized Langevin diffusion using Robbins-Monro step sizes schedules towards the target Gibbs measure. Their framework can cover a variety of numerical discretization schemes, and provides asymptotic analysis under mild assumptions. Authors accomplish this by taking a standard interpolation argument and modifying it to construct a continuous version of the trajectory and verifying the stability conditions from the dynamical systems theory. The scope covers LMC, mirror LMC, proximal LMC, Randomized Mid-Point, and Runge-Kutta methods.

This paper is reviewed by 4 expert reviewers and after the discussion period the paper received the following Rating/Confidence scores: 5/3, 8/3, 7/3, 7/4. All four of the reviewers agree that the paper makes good contributions and it is above the bar. Majority of the reviewers found the contributions significant. AC also agrees with the reviewers and recommends including this paper to conference program.